# Daily soil moisture mapping at 1 km resolution based on SMAP data for desertification areas in Northern China

Pinzeng Rao[1,2], Yicheng Wang[2], Fang Wang[2*], Yang Liu[2], Xiaoya Wang[3], Zhu Wang[2]

[1]State Key Laboratory of Hydroscience and Engineering, Department of Hydraulic Engineering, Tsinghua University, Beijing 100084, China.
[2]State Key Laboratory of Simulation and Regulation of Water Cycle in River Basin, China Institute of Water Resources and Hydropower Research, Beijing 100038, China.
[3]State Key Laboratory of Remote Sensing Science, Faculty of Geographical Science, Beijing Normal University, Beijing 100875, China.
*Correspondence to: Fang Wang (657563390@qq.com)

**Abstract:** Land surface soil moisture (SM) plays a critical role in hydrological processes and terrestrial ecosystems in desertification areas. Passive microwave remote sensing products such as the Soil Moisture Active Passive (SMAP) have been shown to monitor surface soil water well. However, the coarse spatial resolution and lack of full coverage of these products greatly limit their application in areas undergoing desertification. In order to overcome these limitations, a combination of multiple machine learning methods, including multiple linear regression (MLR), support vector regression (SVR), artificial neural networks (ANN), random forest (RF) and extreme gradient boosting (XGB), have been applied to downscale the 36 km SMAP SM products and produce higher spatial-resolution SM data based on related surface variables, such as vegetation index and surface temperature. Desertification areas in Northern China, which are sensitive to SM, were selected as the study area, and the downscaled SM with a resolution of 1 km on a daily scale from 2015 to 2020 was produced. The results showed a good performance compared with in situ observed SM data, with an average unbiased root mean square error value of 0.057 $m^3/m^3$. In addition, their time series were consistent with precipitation and performed better than common gridded SM products. The data can be used to assess soil drought and provide a reference for reversing desertification in the study area. This dataset is freely available at https://doi.org/10.6084/m9.figshare.16430478.v6 (Rao et al., 2022).

**Keywords:** Soil moisture; SMAP; Multiple machine learning; Surface variables; Desertification areas.

## 1. Introduction

Surface soil moisture (SM) plays a very important role in water-energy cycle processes (Sandholt et al., 2002; De Santis et al., 2021) and is an important source of water for plants and soil microbes (Wang et al., 2007; Gu et al., 2008; Mallick et al., 2009). Large-scale areas of northern China are undergoing desertification because of scarce precipitation and insufficient SM. The accurate acquisition of SM is valuable to ecological conservation and revegetation in arid areas of Northern China.

In the past, SM data were mainly obtained through ground measurements or the assimilation of products based on land surface models such as the Global Land Data Assimilation System (GLDAS) (Fang and Lakshmi, 2014; Zawadzki and Kędzior, 2016; Liu et al., 2021). Although most accurate SM data at different soil depths can be obtained, field measurements and in

situ observations are limited due to the high cost and labor intensity involved in their collection and are generally not representative of soil water status over larger areas (Rahimzadeh-Bajgiran et al., 2013; Zhao et al., 2018; Bai et al., 2019). With the development of remote sensing technologies, continuous SM estimates can be generated at regional and global scales (Peng et al., 2021). Compared to ground measurements, remote sensing products can provide good spatial and temporal coverage of SM with a relatively low cost to the user (Zeng et al., 2015; Zhao et al., 2018; Meng et al., 2020). Data assimilation SM products largely depend on the accuracy of the land surface model and the original inputs (Zawadzki and Kędzior, 2016). They generally have low accuracy in areas where ground measurements are scarce, which is a problem that can be overcome with remote sensing.

At present, there are many remotely sensed SM data, some of which are from microwave remote sensing satellites, including active and passive types. SM retrievals from active sensors like Synthetic Aperture Radar (SAR) are sensitive to scattering and greatly affected by the surface roughness and vegetation types (Lievens et al., 2011; Wagner et al., 2013). Unlike active sensors, passive microwave radiometers or sensors are rarely affected by scattering (Abbaszadeh et al., 2019). Common passive microwave SM products are listed in Table 1 below. Studies have compared these products and found that SMAP SM products have higher accuracy and robustness than other remotely sensed SM products (Liu et al., 2019; Wang et al., 2021).

**Table 1: Information of five common passive microwave soil moisture (SM) products.**

| SM Datasets (Abbreviation) | Name | Production source | Resolution | Temporal Coverage | Equator Crossing Time |
|---|---|---|---|---|---|
| AMSR-E/Aqua Daily L3 | Advanced Microwave Scanning Radiometer-Earth Observing System | National Aeronautics and Space Administration (NASA) National Snow and Ice Data Center Distributed Active Archive Center (NSIDC) | 25 km; Daily | 2002-2011 | 1:30 PM Ascending 1:30 AM Descending |
| SMOS | Soil Moisture and Ocean Salinity | European Space Agency (ESA) | 25 km; Daily | 2010-present | 6:00 PM Ascending 6:00 AM Descending |
| FY3B | Fengyun-3B | National Satellite Meteorological Center | 25 km; Daily | 2011-present | 1:40 PM Ascending 1:40 AM Descending |
| GCOM-W1/AMSR2 | Advanced Microwave Scanning Radiometer 2 | Japan Aerospace Exploration Agency (JAXA) | 0.25°/0.1°; Daily | 2012-present | 1:30 PM Ascending 1:30 AM Descending |
| SMAP | Soil Moisture Active Passive | National Aeronautics and Space Administration (NASA) | 36 km; Daily | 2015-present | 6:00 PM Ascending 6:00 AM Descending |

Passive microwave SM products have been applied at watershed and national scale (Fang and Lakshmi, 2014; Meng et al., 2020). However, due to their coarse spatial resolution, microwave SM products have limited applicability to small-scale areas. Compared to microwave sensors, optical satellites such as MODIS and Landsat have a finer spatial resolution. Some observations generated from optical satellites provide good information about SM, such as vegetation index (VI) (usually Normalized Difference Vegetation Index (NDVI) or Enhanced Vegetation Index (EVI)) and land surface temperature (LST)

(Wang et al., 2007; Sun et al., 2012). Many experiments have tried to use these two parameters from optical remote sensing to retrieve surface SM (Mallick et al., 2009; Fang et al., 2013). Based on the LST and VI triangle space, Sandholt et al. (2002) proposed the temperature vegetation dryness index (TVDI) and used it to assess the SM status. Relative SM indicators can be calculated using optical remote sensing data, however, reliable ground measurements or other data are still required to obtain the true value of SM.

Some studies have tried to use surface variables from optical observations to improve the spatial resolution of passive remotely sensed SM products (Peng et al., 2017). Zhao et al. (2017) used the triangle method and Landsat satellite observations to disaggregate coarse-resolution SM data. Studies have shown that polynomial regression is effective in SM and optical observations (Zhao and Li, 2013; Piles et al., 2016). However, these methods have shortcomings in representing the nonlinear relationship between SM and other surface variables (Zhao et al., 2018; Hu et al., 2020). Machine learning methods can be applied to show the nonlinear relationships between SM and surface variables. Random forest (RF) and artificial neural network (ANN) have been widely used in previous studies due to their high generalization ability and robustness (Yao et al., 2017; Liu et al., 2020; Demarchi et al., 2020; Chen et al., 2021). Chen et al. (2021) developed the global surface SM dataset covering 2003–2018 at 0.1° resolution with neural networks and some related variables. Im et al. (2016) used machine learning approaches (RF, boosted regression trees, and Cubist) to downscale AMSR-E SM data in South Korea and Australia and found RF to be superior to the other downscaling methods. Although these machine learning methods perform well in constructing nonlinear regression models, there are still some shortcomings. For example, neural networks are prone to overfitting when the sampling is inefficient (Piotrowski and Napiorkowski, 2013) or variables that are weakly correlated with the dependent variable (Elshorbagy and Parasuraman, 2008; Ågren et al., 2021). Since the RF algorithm uses random sampling with replacement, its simulation results will not exceed the range of training set and tend to ignore some extreme values when used as a regression model (Belgiu and Drăguţ, 2016). Additionally, it does not perform well when learning from an extremely imbalanced training data (Lin et al., 2021). Extreme gradient boosting (XGB), as a new ensemble learning method (Chen and Guestrin, 2016), performs well in some fields (Wang et al., 2020; Fan et al., 2021; Ma et al., 2021), but it has rarely been used for soil moisture downscaling. Compared with methods such as RF, the XGB algorithm adopts the boosting weighted sampling method, which can reduce the impact of imbalanced data and better simulate the extreme values existing in the samples (Chen and Guestrin, 2016). The coarse-resolution remote sensed SM (>10 km) itself has ignored some maxima or minima with relatively finer-grid SMs, so a method that better simulates extreme values will obviously have certain theoretical advantages.

The selection of feature variables is critical for regression models. In addition to LST and VI mentioned above, variables such as terrain and soil conditions also have a significant impact on SM. Abbaszadeh et al. (2019) downscaled SMAP radiometer SM products over the continental United States using MODIS products (including NDVI and LST), precipitation and topographic data, and evaluated the influence of soil texture on SM. Zhao et al. (2018) added additional surface variables, such as leaf area index (LAI), normalized difference water index (NDWI), surface albedo and the solar zenith angle. Hu et al. (2020) added the normalized shortwave-infrared difference bare soil moisture index (NSDSI), horizontally polarized

Brightness Temperature (TBh) and vertically polarized Brightness Temperature (TBv) to the regression model. In general, all
these variables can be classified into vegetation, temperature, soil wetness, topography, and soil factors and sensors conditions.
In recent years, the Chinese government has carried out afforestation activities in order to reverse desertification in the
North. Considering the role of SM in terrestrial ecosystems, it is urgent to obtain accurate SM with high temporal and spatial
resolution. This study aims to downscale SMAP SM products by constructing a nonlinear relationship between SM and related
surface variables by means of multiple machine learning methods and generate SM products with higher temporal and spatial
resolution in desertification areas. The in situ observed SM data from the Maqu Monitoring Network and Babao Monitoring
Network, several gridded SM products, and precipitation and temperature data from meteorological stations were used for
validation and analysis.
**2. Materials and methodology**
**2.1 Study area**
Northern China is mostly arid with an annual precipitation of generally less than 600 mm, and is subject to large-scale
desertification. The desert areas of Northern China are susceptible to climate and hydrological changes and have fragile
ecosystems. Soil water is a key parameter in land-atmosphere interactions (Ma et al., 2019), and its change greatly affects the
survival of vegetation and agricultural production in desertification areas. The studied area whose boundaries are provided by
government departments used for this study covers 3.36 million km$^2$, encompassing seven provinces (Fig. 1). The precipitation
in the study area decreases gradually from southeast to northwest, and belongs to the temperate continental climate (Fig. 1).
The terrain is complex, and the average elevation is approximately 1,900 m, ranging from -192 m to 7,439 m.

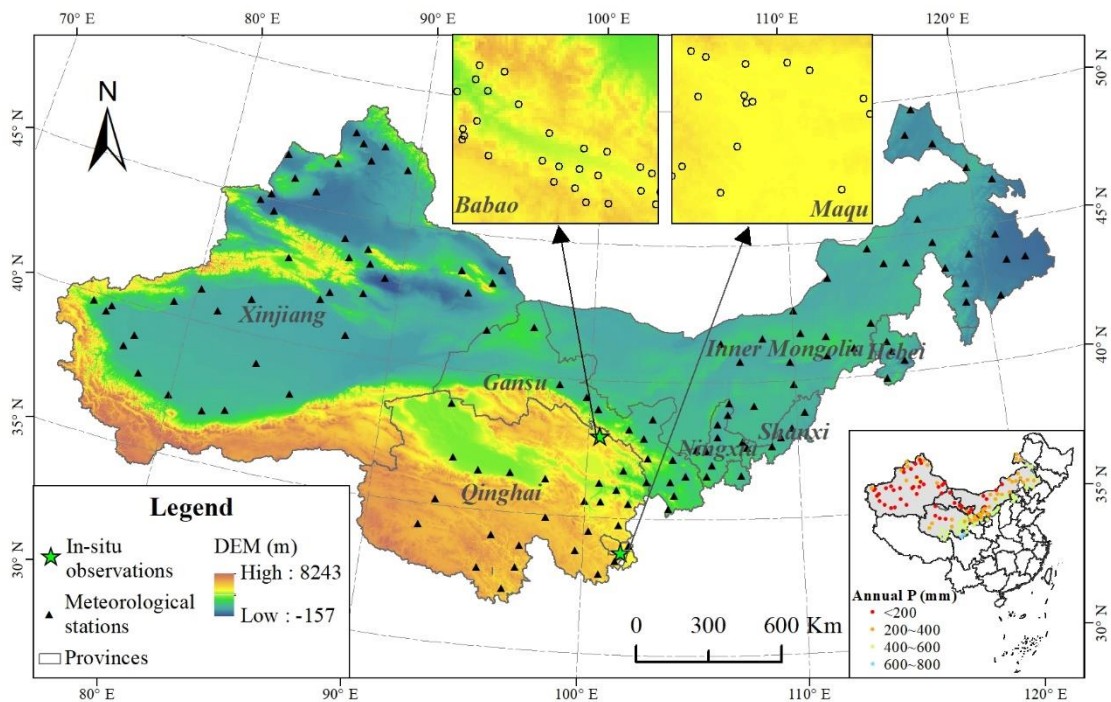

**Figure 1: Location of the study area.**

## 2.2 Observations for the production of soil moisture data

### 2.2.1 SMAP SM data

The SMAP satellite was launched on January 31, 2015. Its mission consists of an L-band radar and radiometer instrument suite, which provides global measurements and monitoring of SM in the top 5 cm of soil. The Level-3 products are daily composites of the Level-2 products and are the most commonly used for applications. The Level-3 products are available in three spatial resolutions: 36 km passive, 9 km active-passive, and 3 km active (O'Neill et al., 2010). Following the malfunctioning of its radar in 2015, SMAP radar data were replaced with those of Sentinel-1, limiting the application of active and active-passive products.

The SMAP Level-3 passive daily SM product (L3_SM_P, Version 6) with a grid resolution of 36 km has been produced since March 31, 2015. Zeng et al. (2015) showed that most of remotely sensed SM products were slightly better during daytime than during nighttime, and the same conclusion for the SMAP SM product was confirmed by Zhao et al. (2018). Therefore, the SMAP Level-3 SM product with the descending overpass time of 6:00 AM was used in this study. The data were downloaded from NASA Earthdata (https://search.earthdata.nasa.gov).

### 2.2.2 MODIS products

MODIS provides continuous time-series predictors for important parameters, such as vegetation index and surface temperature. This paper used MODIS products MOD09A1, MOD11A1, MOD13A2, MOD15A2H and MCD43D58 (Table 2). The 1-km daily LST was provided by MOD11A1, and the 1-km 16-day EVI and NDVI was provided by MOD13A2. MOD15A2H provided 8-day Leaf Area Index (LAI) with a spatial resolution of 500 m. MCD43D58 provided daily albedo data with a spatial resolution of 30 arc second (~1,000 m). Some soil wetness related indexes, including NDWI, NSDSI, and Land Surface Water Index (LSWI), were produced by MOD09A1. Their formulas are:

$$NDWI = (B_4 - B_2)/(B_4 + B_2) \tag{1}$$

$$LSWI = (B_2 - B_6)/(B_2 + B_6) \tag{2}$$

$$NSDSI = (B_6 - B_7)/B_6 \tag{3}$$

where $B_2$, $B_4$, $B_6$ and $B_7$ represent the MOD09A1 surface reflectance of the 2nd, 4th, 6th and 7th bands, respectively.

These MODIS products are available from NASA Earthdata (https://search.earthdata.nasa.gov). All data were obtained from 2015 to 2020 and processed to a spatial resolution of 1,000 meters.

**2.2.3 Topographic data**

Topographic factors are strongly related to SM, including elevation, slope and aspect (Bai et al., 2019; De Santis et al., 2021). The Shuttle Radar Topography Mission (SRTM) digital elevation model (DEM) at 3 arc second resolution (~100 m), version 3, obtained from the Land Processes Distributed Active Archive Center (LP DAAC)( https://e4ftl01.cr.usgs.gov/MEASURES/SRTMGL3.003/), was used as elevation. Slope and aspect were generated based on the DEM.

**2.2.4 Soil texture data**

Soil texture, the proportions of sand, silt and clay particles, controls the water holding capacity of the soil. The soil data at 1,000 m resolution, including the proportions of sand, silt and clay, used for this study used the China Soil Characteristics Dataset (CSCD) (Shangguan et al., 2012), obtained from National Tibetan Plateau Data Center (http://westdc.westgis.ac.cn/).

**2.2.5 In Situ SM observations**

The in situ SM measurements were collected from the data provided by the Maqu Monitoring Network (Zhang et al., 2020) and the Babao Monitoring Network (Kang et al., 2017). The Maqu Monitoring Network covers 26 sites and provides SM for the surface layer (0-5 cm) at 15-minute intervals from 2009 to 2019; 19 of the available sites which have data after 2015 were used in this study (Fig. 1). The Babao Monitoring Network covers 40 sites and provides hourly SM for the surface layer (4 cm, 10 cm and 20 cm) from 2013 to 2017; 29 of the available sites have data after 2015 and their observations of the first layer (4 cm) were used in this study (Fig. 1). To compare with the simulated results, they were all processed into daily time series.

## 2.2.6 Precipitation and temperature data

The daily precipitation and temperature data were acquired from 131 meteorological stations from the China Meteorological Data Service Centre (http://data.cma.cn). The spatial locations of these meteorological stations are shown in Fig. 1. The average annual precipitation of most stations from 2015 to 2020 is less than 600 mm, and gradually decreases from northwest to southeast (Fig. 1).

**Table 2: Main predictors used in the study and corresponding datasets**

| Datasets | Predictors | Original spatial resolution | Temporal resolution | Number of granules (Years×tiles) |
|---|---|---|---|---|
| SMAP | SM | ~36 km | Daily | 2,064 |
| MOD11A1 | LST | 1 km | Daily | 17,460 |
| MOD13A2 | NDVI; EVI | 1 km | 16-day | 1,104 |
| MOD15A2H | LAI; FAPAR | 500 m | 8-day | 2,208 |
| MOD09A1 | NDWI; LSWI; NSDSI | 500 m | 8-day | 2,208 |
| MCD43D58 | Albedo | 30 arc second (~1 km) | Daily | 2,192 |
| SRTM | DEM; Slope; Aspect | 3 arc second (~100 m) | - | 32 |
| CSCD | Sand; Silt; Clay | 1 km | - | 1 |

## 2.2.7 Other gridded SM datasets

Some other gridded SM data were used to compare the simulation results (Table 3). The SMAP Level-2 product (L2_SM_SP) merges SMAP radiometer and processed Sentinel-1A/1B SAR observations. It is available at 3 km and 1 km resolutions. The Global Change Observation Mission Water (GCOM-W1) AMSR2 product is produced by the Japan Aerospace Exploration Agency (JAXA), and SM data at a 0.1° spatial resolution were selected for this study. The Copernicus Climate Change Service (C3S) produces a global SM gridded dataset from 1978 to present from satellite sensors such as SMOS, AMSR2 and SMAP. It has a spatial resolution of 0.25 degrees and offers three types of products: active, passive and combined. The combined product that we used in this study is generated by merging the active and passive products. The fifth generation reanalysis dataset (ERA5) produced by European Centre for Medium-Range Weather Forecasts (ECMWF) provides several variables including volumetric soil water over several decades. In this dataset, the soil is divided into four layers and the depth of the top layer is 0-7 cm. In this study, we downloaded the hourly volumetric soil water data of the top layer and processed them as daily averages. Famine Early Warning Systems Network (FEWS NET) Land Data Assimilation System (FLDAS) provides daily SM at a 0.01° spatial resolution over the Central Asia region (30-100° E, 21-56° N), which covers part of our study area. The product consists of four layers of SM, and the SM at the top layer (0-10 cm) was selected for this study.

**Table 3: The gridded SM products used in this study**

| Institution | Name | Soil layers | TYPES | Temporal resolution | Grid spacing | Data link |
|---|---|---|---|---|---|---|
| NASA | SMAP/ Sentinel-1 (L2_SM_SP) | One layer (0-5 cm) | Active microwave | 1-2 days | 1/3 km | https://cmr.earthdata.nasa.gov/search/concepts/C1931663473-NSIDC_ECS.html |
| JAXA | GCOM-W1/AMSR2 | One layer (~) | Passive microwave | Daily | 0.1°/0.25° (~11 km/28 km) | https://gportal.jaxa.jp/gpr/ |
| ECMWF | C3S | One layer (~) | Passive, active and combined | Daily | 0.25° (~28 km) | https://cds.climate.copernicus.eu/cdsapp#!/dataset/satellite-soil-moisture |

| ECMWF | ERA5 | Four layers (0-7 cm, 7-28 cm, 28-100 cm,100-289 cm) | Reanalysis | Hourly | 0.1°(~11 km) | https://cds.climate.copernicus.eu/cdsapp#!/dataset/reanalysis-era5-land |
|---|---|---|---|---|---|---|
| NASA | FLDAS | Four layers (0-10 cm, 10-40 cm, 40-100 cm,100-200 cm) | Reanalysis | Daily | 0.01°(~1.1 km) | https://cmr.earthdata.nasa.gov/search/concepts/C2020764153-GES_DISC.html |

**2.3 Downscaling approach based on multi-machine learning**

According to the selected variable indicators (mainly including topographic data, soil data and some MODIS products) and machine learning methods, we constructed a framework to downscale SMAP SM based on multiple machine learning methods (Fig. 2).

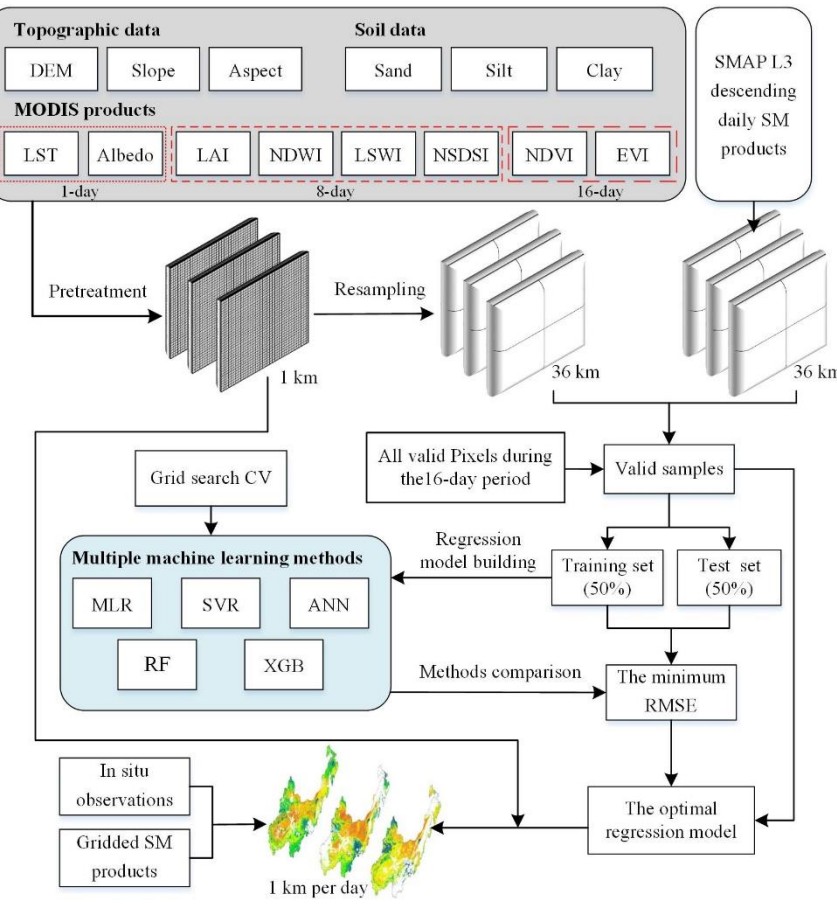

**Figure 2: Schematic of the SMAP soil moisture downscaling framework**

### 2.3.1 Machine learning methods

Machine learning methods are widely used in regression and classification. We selected machine learning methods that are currently widely used to build regression models for SM and its related variables. We studied five methods: multiple linear regression (MLR), support vector regression (SVR), artificial neural networks (ANN), random forest (RF) and extreme gradient boosting (XGB). MLR and SVR have been widely used as regression methods in the past (Yu et al., 2012; Achieng, 2019; Wang et al., 2019). ANN is currently one of the most popular machine learning methods and is used in many fields, including the inversion of remotely sensed SM (Del Frate et al., 2003; Elshorbagy and Parasuraman, 2008; Yao et al., 2017; Chen et al., 2021).

RF and XGB are tree based ensemble algorithms, which have prediction accuracy and good generalization ability, and are not prone to overfitting (Rao et al., 2018; Abbaszadeh et al., 2019). RF is a multiple-tree algorithm improved by bootstrap to reduce decision tree bias in determining the splits (Mohana et al., 2021). Several studies have used RF to build regression models of remotely sensed SM and related variables, and almost all achieved better results compared to other regression methods (Zhao et al., 2018; Qu et al., 2019; Hu et al., 2020). In contrast, the application of XGB, which applies a regularized gradient boosting framework, is still very limited. However, XGB has prominent advantages in generalization performance and accuracy (Wang et al., 2020).

The XGB algorithm is a boosting-type ensemble of multiple CART decision trees (Chen and Guestrin, 2016). The predicted result of the boosting-type tree ensemble model can be expressed as follows:

$$\widehat{y_i} = \emptyset(x_i) = \sum_{k=1}^{K} f_k(x_i), f_k \in F \tag{4}$$

where $F$ is the space of regression tree, $K$ is the total number of trees, which means the model uses $K$ additive functions, $f_k(x_i)$ is the weighted score of the $k$-th tree on $i$-th input data $(x_i)$.

XGB adopts a regularized learning objective to optimize the simulation results.

$$Obj(\emptyset) = \sum_{i=1}^{N} l(y_i, \widehat{y_i}) + \sum_{k=1}^{K} \Omega(f_k) \tag{5}$$

where $l$ is the loss function, $N$ is the total number of input, $\Omega$ is the regularization term to penalize the model complexity and prevent overfitting.

Compared with RF and other some methods, XGB has significantly faster calculation speed (Fan et al., 2018; Shi et al., 2021). Some studies have shown that XGB is a better regression and classification algorithm than RF and other machine learning methods (Ågren et al., 2021; Fan et al., 2021).

### 2.3.2 The construction of 16-day regression model

The downscaling process is shown in Fig. 2. First, all data need to be preprocessed. Daily LST data are likely to be affected by the cloud, so we performed quality control to MOD11A1 products using its quality control (QC) band and choose high-quality cloud-free pixels. All selected variables, including LST, Albedo, LAI, NDWI, LSWI, NSDSI, NDVI, EVI, DEM,

slope, aspect, sand, silt and clay, were aggregated into a resolution of 1 km with a geotiff format. These variables were further resampled to the spatial resolution of the SMAP SM data (36 km) using the nearest neighbor interpolation method.

Second, valid samples were obtained and split. Since it is severely affected by noise (such as clouds), MOD11A1 only provides daily valid clear-sky LST values onto grids. In addition, each SMAP image has a narrow coverage and provides only a small number of valid pixels per day. It means that there may be few or no valid samples if only the data of a certain day are selected to build the regression. The variables from MOD13A2 and MOD15A2H are the best composite within 16 days and 8 days, respectively. To overcome the limitation, we chose to build regression models within 16 day periods (the lowest temporal resolution from these dynamic variables). All valid data (including training and test sets) within 16 days were used as the samples in the regression model. For instance, for NDVI and EVI on January 1, 2020, which are composite results from January 1st to 15th, the valid data during the period were used as samples. The number of valid samples for surface variables and SMAP SM for each period in 2015-2020 is shown in Fig. 3. The day of year (DOY) is used to represent the corresponding period. Since limited available SMAP SM grid data, there may be few valid samples we can obtain during cold seasons. The valid samples for each period were divided into training and test sets, each accounting for 50% of the total number of samples. In this study, stratified random sampling based on sampling date during the 16-day period was employed to split the training and test sets. Moreover, to avoid excessively inconsistent training and test sets, the Kolmogorov-Smirnov (KS) test is adopted to test the distribution consistency of them (Kovalev and Utkin, 2020). If the p-value of the KS test result is less than or equal to 0.05, stratified random sampling is performed again, and until the requirements are met.

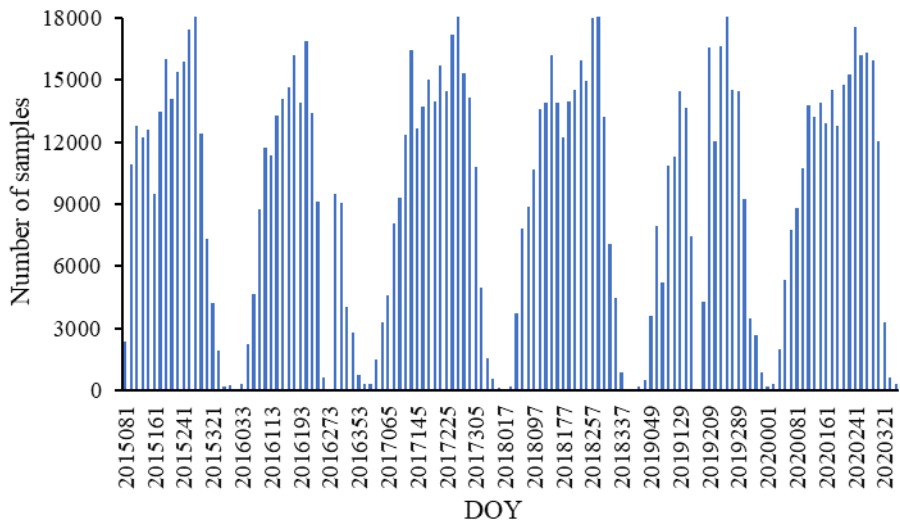

**Figure 3: The number of valid samples for a 16-day period in 2015-2020. DOY is the day of year, the same below.**

Third, the regression model was determined based on training and test sets. Considering the number of samples is critical to the accuracy of the regression model, we only selected periods with more than 100 samples to build the model and DOY of 2016017, 2018017, 2018353, 2019001 and 2019177 were excluded. Then, we used the training set and multiple machine

learning methods (MLR, SVR, ANN, RF and XGB) to build a regression model for each 16-day period. The regression model was then defined according to the selected machine learning method:

$$SM = f(LST, Albedo, LAI, NDWI, LSWI, NSDSI, NDVI, EVI,$$
$$DEM, slope, aspect, sand, silt \ and \ clay) \tag{6}$$

where $f$ represents the regression function of the machine learning method (MLR, SVR, ANN, RF or XGB).

Finally, hyperparameter turning and the selection of the optimal model. Hyperparameters are critical for some machine learning methods (Klein et al., 2017; Khan et al., 2020; Sun et al., 2021). In this study, the key hyperparameters of SVR, ANN, RF and XGB are tuned based on grid search cross-validation (CV). All models are evaluated based on the correlation coefficient (R) and the root mean square error (RMSE). They are calculated as follows:

$$R = \frac{Cov(SM_I, SM_P)}{\sqrt{Var(SM_I)Var(SM_P)}} \tag{7}$$

$$RMSE = \sqrt{\frac{1}{n}(SM_P - SM_I)^2} \tag{8}$$

where $SM_I$ is the SMAP SM, $SM_P$ is the corresponding SM predicted by the regression model, *Cov* represents the covariance function, *Var* is the variance, and $n$ is the number of valid samples for $SM_I$ or $SM_P$.

The RMSE is used as the evaluation metric for hyperparameter turning. The tuning results of hyperparameters are shown in Tables S2 and S3. According to the optimal hyperparameter, the corresponding model can be constructed.

### 2.3.3 Prediction of 1-km daily SM product

The accuracy of the five regression models was compared using the average RMSE of training and test sets. This average RMSE can be expressed as:

$$\overline{RMSE} = \frac{RMSE_{Training} + RMSE_{Test}}{2} \tag{9}$$

where $RMSE_{Training}$ and $RMSE_{Test}$ are the RMSE of training and test sets for these models, respectively.

The regression model with the smallest $\overline{RMSE}$ was selected as the optimal model. Furthermore, we used the selected optimal model and these surface variables with a resolution of 1 km within 16 days to simulate daily SM at 1 km resolution on the corresponding date. Taking 16 days as a period, all daily SM data with a spatial resolution of 1 km from 2015 to 2020 were predicted. In addition, to obtain a more complete time series of SM data, we used the model of the previous period when the number of valid samples was less than 100.

### 2.4 Evaluation method

The in situ SM measurements were used to validate the downscaled results. In addition to R and RMSE, bias and unbiased RMSE (ubRMSE) were also used for accuracy evaluation. Bias indicates the overall level of overestimation or underestimation of simulation results. ubRMSE can eliminate the influence of deviation. They were calculated according to:

$$ubRMSE = \sqrt{\frac{1}{n}((SM_{In} - \overline{SM_{In}}) - (SM_D - \overline{SM_D}))^2} \qquad (10)$$

$$bias = \overline{SM_{In}} - \overline{SM_D} \qquad (11)$$

where $SM_{In}$ is the in situ observed SM, $SM_d$ is the downscaled SM of the corresponding grid, and $n$ is the number of valid
samples for $SM_{In}$ or $SM_D$.

## 3. Results

### 3.1 Model comparison

The daily SM data from DOY 81 in 2015 to DOY 366 in 2020 were simulated producing 128 regression results every 16

days. The correlation coefficient (R) and the root mean square error (RMSE) of each regression result for the training set and
the test set are shown in Fig. 4 and Fig. 5, respectively. According to Equation 9, among the 128 regression results, there were
114 from the XGB model, and 14 from RF.

For all models except MLR, R is greater than 0.6 and RMSE is less than 0.05 $m^3/m^3$ both for training and test sets. R

values greater than 0.6 and 0.8 indicate reliable and strong correlations, respectively (Akoglu, 2018). It means that all methods
except MLR have reliable simulation accuracy. For the training set using XGB, Rs are all above 0.96, generally higher than
for other methods; Similarly, the RMSEs of XGB are all lower than 0.02 $m^3/m^3$, generally lower than those of other methods.
The R of RF is second only to that of XGB, and for several periods it is higher than for XGB; the RMSEs of RF are also
generally lower than 0.02 $m^3/m^3$ and are lower than those of XGB in several periods. SVR and ANN perform generally better
in the cold season, and worse in other seasons. In general, their results are inferior to those of XGB and RF. The simulation
results of MLR are relatively poor both in terms of RMSE and R.

The results of the test set show that XGB, RF and SVR perform better than ANN and MLR. Table 4 shows the average

RMSE and R values of the training and test sets over all periods, and the performance order of the model can be obtained as
XGB>RF>SVR >ANN >MLR. In addition, there are seasonal variations in R and RMSE both for training and test sets.
Moreover, the evaluation accuracy is generally better in the cold season, when sample sizes are relatively small.

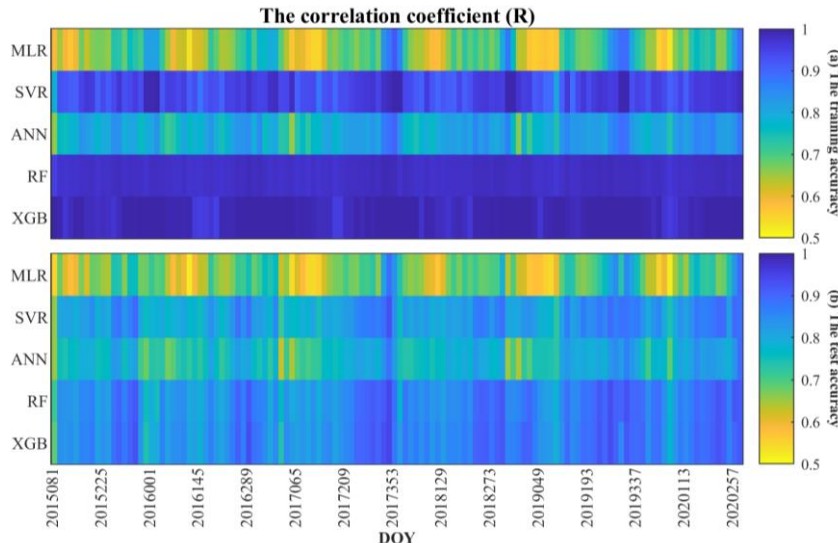


Figure 4: The correlation coefficient (R) of the models (MLR, SVR, ANN, RF and XGB) on different periods: (a) The training

accuracy; (b) The test accuracy.

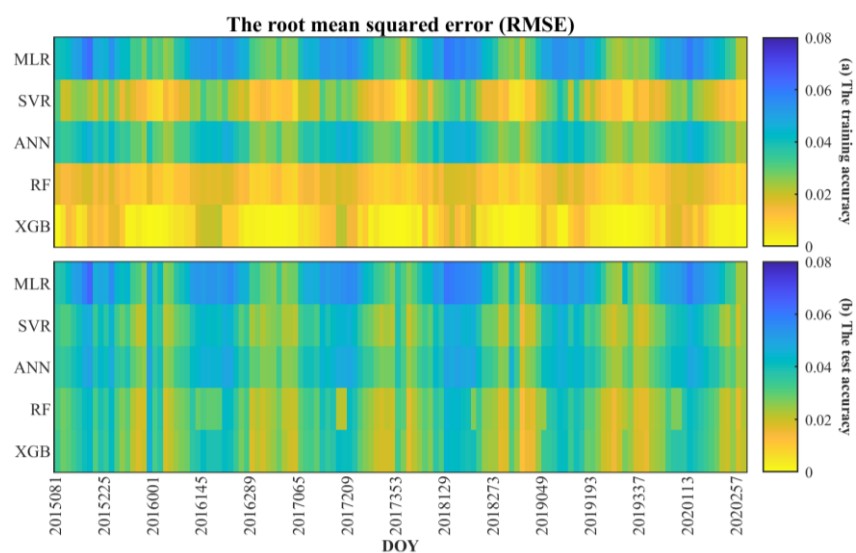


Figure 5: The root mean square error (RMSE) of the models (MLR, SVR, ANN, RF and XGB) for different periods: (a) The training

accuracy; (b) The test accuracy.

Table 4: Accuracy of the models based on correlation coefficient (R) and root mean square error (RMSE)

| | Model | MLR | SVR | ANN | RF | XGB | Combination |
|---|---|---|---|---|---|---|---|
| Training set | R | 0.688 | 0.943 | 0.864 | 0.978 | 0.991 | 0.992 |
| | RMSE ($m^3/m^3$) | 0.042 | 0.019 | 0.028 | 0.013 | 0.007 | 0.007 |
| Test set | R | 0.675 | 0.824 | 0.660 | 0.857 | 0.861 | 0.861 |
| | RMSE ($m^3/m^3$) | 0.043 | 0.033 | 0.047 | 0.030 | 0.029 | 0.028 |

**3.2 Comparison with the in situ data and precipitation**
The downscaled 1 km gridded SM were compared with the in situ SM observations of the Maqu Network and Babao
Network (Fig. 6). Due to the difference in sensors, soil depth and measurement scale (point observation in case of the in situ
measured SM and 1 km grid for the downscaled SM), there is a certain deviation between in situ observation data and the
downscaled gridded SM data. The downscaled SM of most sites at the Maqu Network and Babao Network are highly correlated
with the in situ measured SM (R>0.6). In the Maqu Network, the ubRMSEs with an average of 0.057 $m^3/m^3$ are all less than
0.090 $m^3/m^3$, and the bias ranges from -0.10 to 0.22 $m^3/m^3$. In the Babao Network, the average ubRMSE of all sites is 0.081
$m^3/m^3$, and some of them exceed 0.1 $m^3/m^3$. In addition, their bias ranges from -0.07 to 0.45 $m^3/m^3$. It means that the validation
accuracy of Babao Network is generally lower than Maqu Network. That may be mainly because the measured soil depth at
the Babao Network is 4 cm, which means that there could be a systematic error between the datasets. Therefore, the validation
accuracy should mainly refer to the evaluation accuracy of Maqu network.

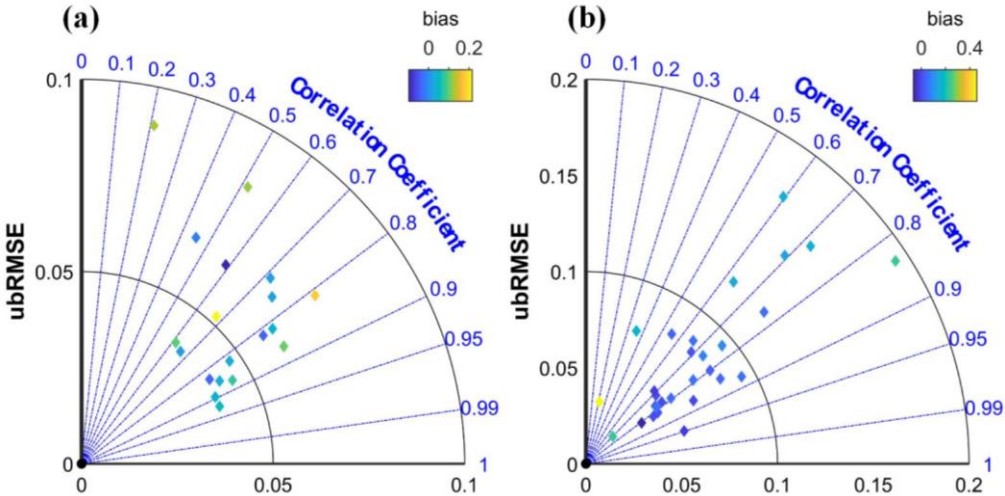

**Figure 6: The relationships between in situ SM and downscaled SM. (a) Maqu Network; (b) Babao Network.**
To better understand the reason for these poor results, the scatter plots comparing the two sets of data were drawn. Figure
7 shows the results of the 19 sites of the Maqu Network. All four statistical metrics, namely, R, RMSE, ubRMSE and bias
were calculated, and their fitting line of the scatter was plotted. Not surprisingly, the relationship is generally improved where
there are more valid data. It means that the validation effect of in situ observations is affected by the amount of data. The same
conclusion can be drawn through 29 sites at the Babao Network (Fig. S1).

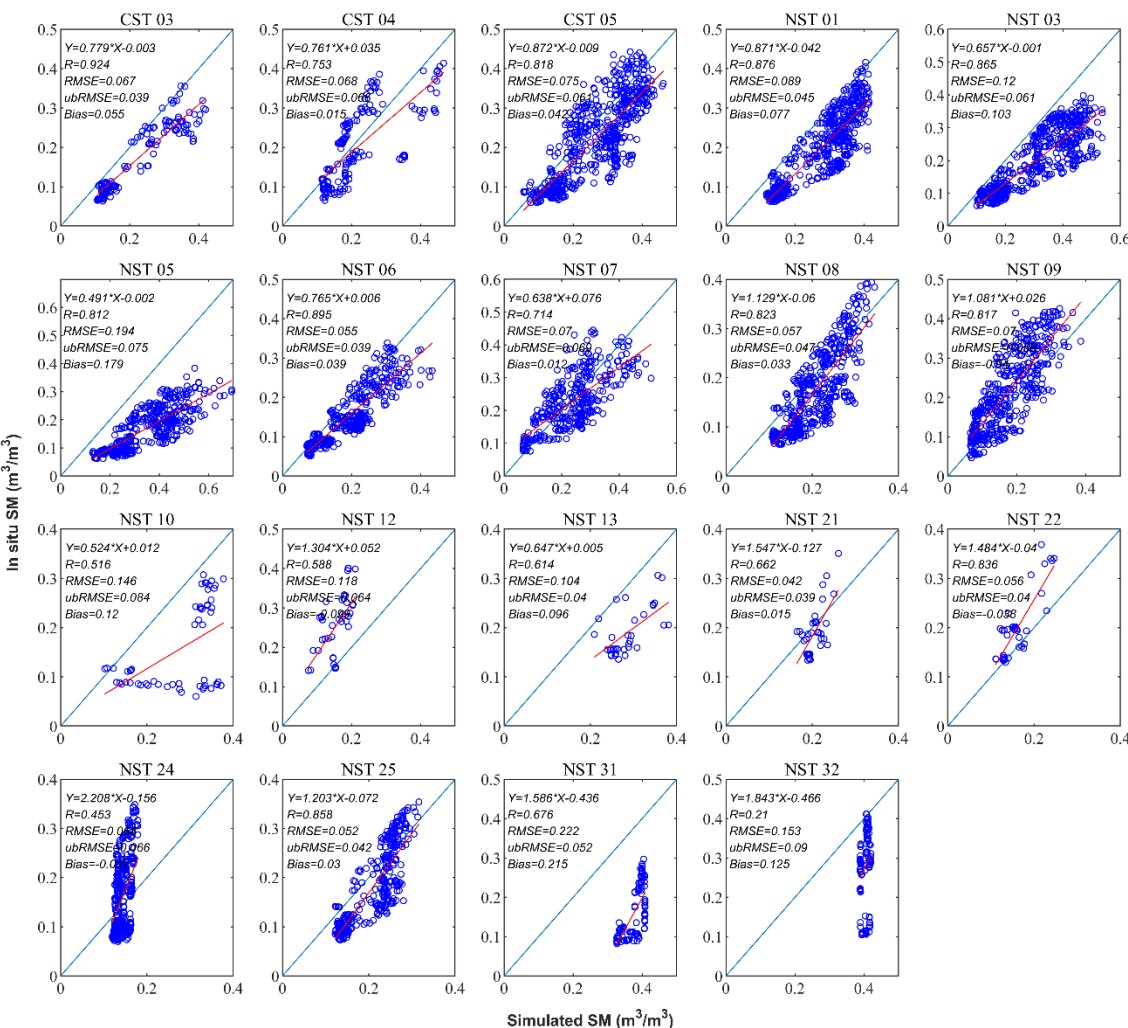

**Figure 7: Comparison between the downscaled SM and in situ SM of the Maqu Network.**

All SM products are compared with in situ SM. Figure 8 shows a significantly higher correlation between the downscaled SM and in situ SM of the Maqu Network. The median ubRMSE of the downscaled SM is the smallest, and its RMSE is second only to the C3S (0.25°) product. The bias of the downscaled SM is higher than that of some products, even higher than the original SMAP L3 (36 km) data. Almost the same results can be obtained from in situ observations of Babao Network (Fig. S2). The difference is that the bias of the downscaled SM is lower than the result of SMAP L3 (36 km). Compared with the RF-based and the XGB-based downscaled SMs, the downscaled SM with multiple machine learning approaches performed better, especially R and ubRMSE.

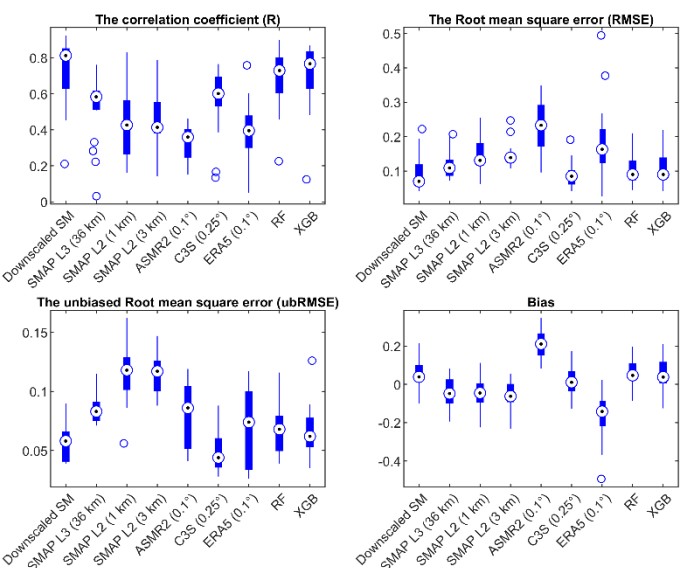

**Figure 8: Comparison of gridded products and in situ observation SM of the Maqu Network.**

The observed SM of sites with a greater number of observed data were compared with these gridded SM data at different resolutions and precipitation. Figure 9 shows the temporal variations of these SM at four sites 2016-2017. The relationship between in situ observed SM and precipitation at all four sites is very consistent, showing annual fluctuation. The greater SM corresponds to more precipitation during the hot season, and the smaller SM corresponds to less precipitation during the cold season.

Except for GCOMW/ASMR2 SM, the variation trends of these acquired gridded SM and the downscaled SM are basically the same despite the large difference in spatial resolution. GCOMW/ASMR2 significantly undestimates SM compared to other products. Both the SMAP L2 SM at 1 km and 3 km may be overestimated (CST05) and may also be underestimated (WSN18) compared with in situ observations. Moreover, SMAP L2 SM has some valid data mainly on hot days and almost no valid data during cold seasons. The peak values of the ERA5 SM are close to those of the in situ observations, but the low values are overestimated. The C3S SM is similar to the 36 km SMAP SM, and its peak values are simulated more accurately, while the minimum values have little valid data. Compared with the original data (36 km SMAP L3), the downscaled SM has a more complete time series, especially during the cold season. The downscaled SM data almost all match well with the in situ measured SM data, and all of them are consistent with the precipitation. The difference between the downscaled SM and the in situ measured SM is mainly reflected in the magnitude of the variation, which is probably due to the difference in spatial resolution.

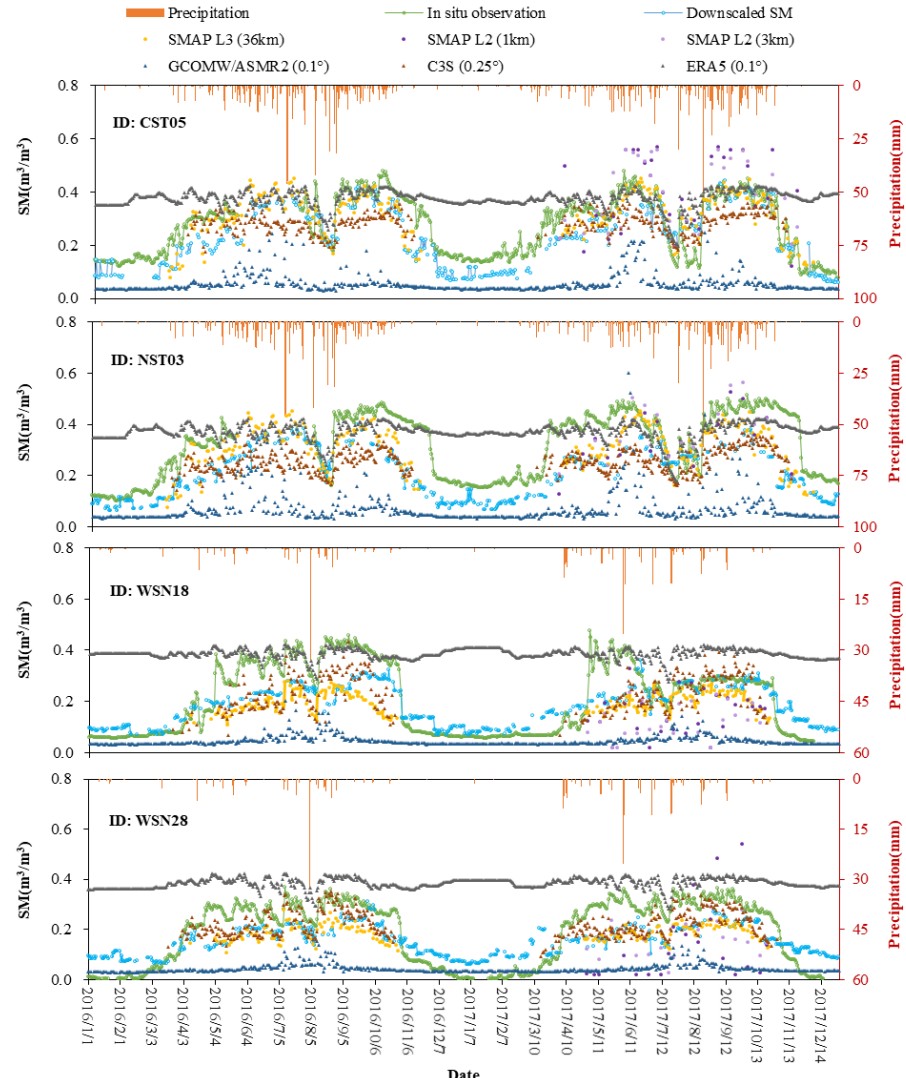

**Figure 9: Time series of the in situ observed SM, the downscaled SM, the acquired gridded SM products and daily precipitation at the four selected SM sites (From Maqu Network and Babao Network, respectively) in 2016-2017.**

**3.3 Mapping of the downscaled SM**

SM varies greatly in different months in desertified areas. Figure 10 shows the average SM in each month in the study area. The SM shows a monthly change pattern, and the values from June to September are bigger than in other months, especially in southern Qinghai Province, eastern Inner Mongolia Province, and western Xinjiang Province. The SM in some areas is low throughout the year, such as in the Tarim Basin of Xinjiang Province, western Inner Mongolia Province and most of Gansu Province.

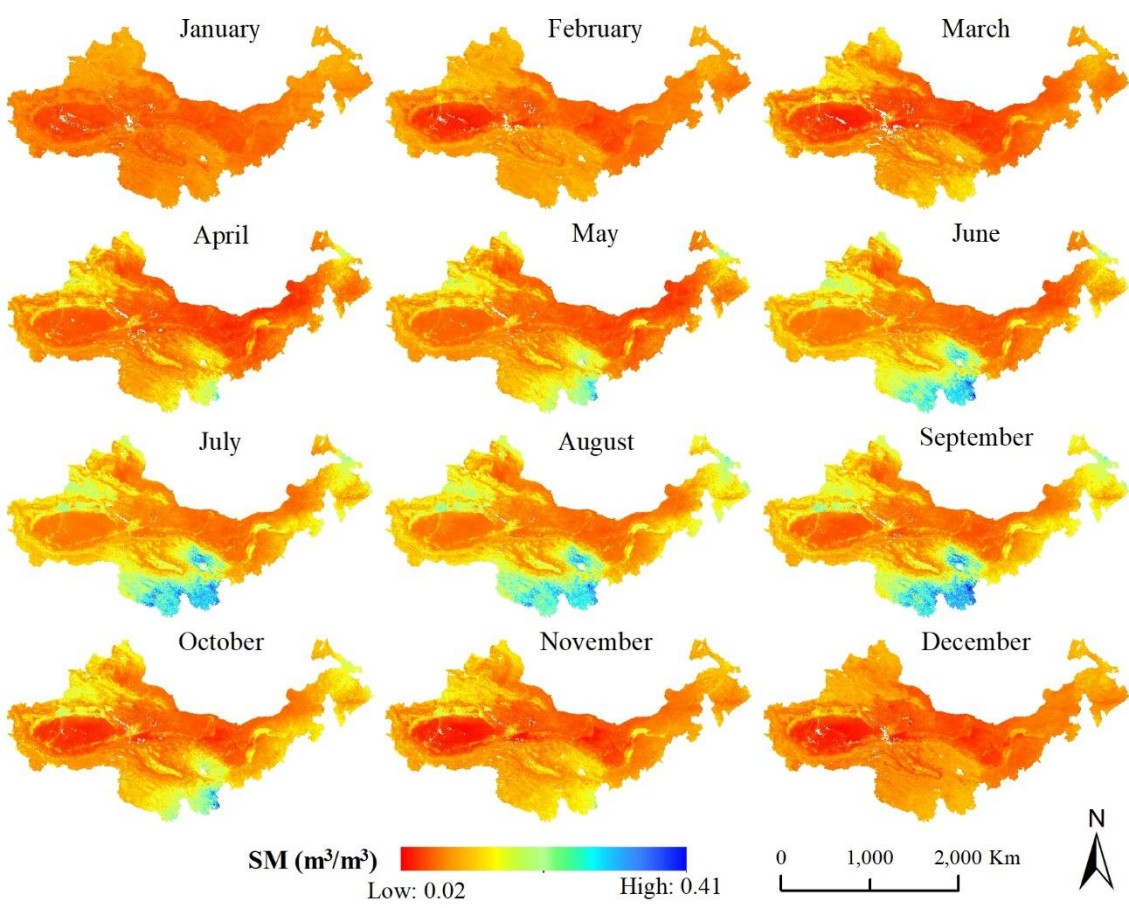

**Figure 10: Monthly average SM in the study area.**
The annual average SM was calculated (Fig. S3). Compared with the monthly average SM, the annual average SM
changed significantly less. Further, we compared the spatial patterns of the downscaled SM with the gridded SM products with
different resolutions. Figure 11 shows the daily average SM of these products from 2015 to 2020. The spatial patterns of the
downscaled SM and 36 km SMAP SM are basically consistent, but the downscaled data show better details in some areas such
as near rivers. The overall values of GCOMW SM are relatively small, and exhibit some obvious errors in some areas. For
example, SM in the Tarim Basin is higher than in the surrounding area, which is completely inconsistent with other SM data.
The spatial pattern of the C3S SM is close to the downscaled SM and the 36 km SMAP SM, but some details are not presented.
For example, SM in the Hetao Plain along the Yellow River is much higher than that in its surrounding area, which can be
found in the downscaled SM and the SMAP SM, but not in the C3S SM. There are obvious errors in the results of ERA5. The
average SM is significantly overestimated in the southern part of the study area, and underestimated in some areas in the
northern of the study area. The FLDAS SM has high resolution, and its overall spatial pattern is relatively consistent with the
downscaled SM and 36 km SMAP SM. The difference is that the FLDAS SM is significantly larger in higher elevation areas
of the west than in other regions, which is quite different from other products. This suggests that the FLDAS SM may be

overestimated in these regions. In addition, FLDAS SM does not show wetter soil along the river. The spatial patterns of the RF-based and XGB-based downscaled SMs are both close to that of the downscaled SM with multiple machine learning approaches, however, the the maximum SM based on RF is smaller than the results based on XGB and multi-model combination.

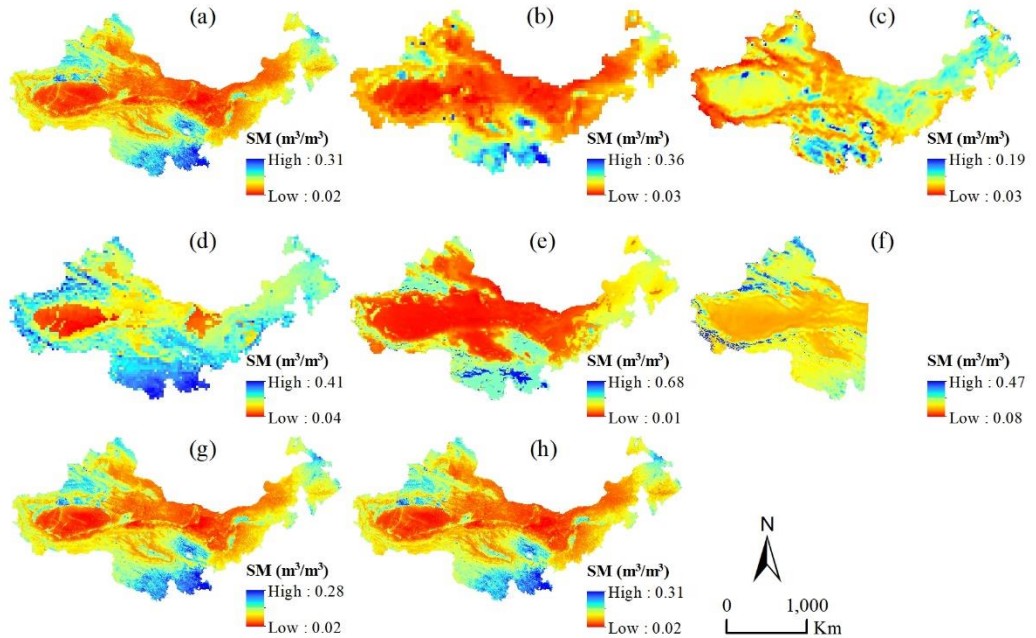

**Figure 11: Daily average SM from 2015-2020 in the study area. (a)-(f) are the downscaled SM (1 km), SMAP L3 SM (36 km), GCOMW/ASMR2 SM (0.1°), C3S SM (0.25°), ERA5 SM (0.1°), FLDAS SM (0.1°), RF-based and XGB-based downscaled SMs (1 km), respectively.**

To better demonstrate the differences in SM, a case of the Mu Us Desert was selected (Fig. 12). The Mu Us Desert is located in a semi-arid area with annual average precipitation of less than 400 mm, decreasing gradually from southeast to northwest. The main types of land cover are grassland and sandy land, and the salinization is serious in a few areas. Desertification has been severe for a long time in the past but significantly reversed with artificial afforestation in recent years.

SM shows an overall trend of gradual decrease from the southeast to the northwest (Fig. 12 (b)~(g)), which is consistent with the distribution of precipitation. The average SM of the same location changes little from year to year. Overall, it is relatively large in 2018 and relatively small in 2015, which is roughly consistent with annual precipitation patterns. Land cover types also have a certain influence on the spatial difference of SM. The northwestern portion of the Mu Us Desert is mainly grassland, which is strongly dependent on precipitation (Fig. 12 (h)). The southeastern area is mainly cultivated land and is less affected by precipitation as it relies on pumping groundwater rather than natural precipitation (Fig. 12 (j)).

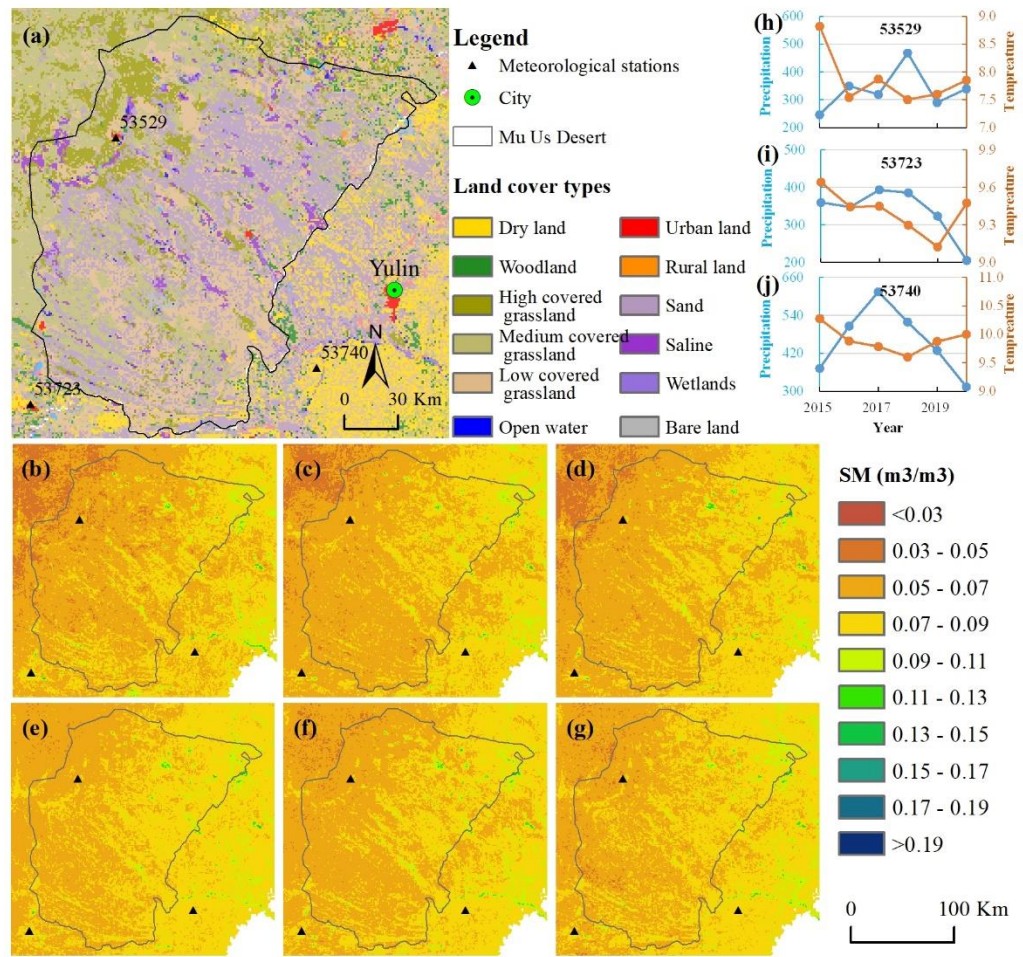

Figure 12: Soil moisture estimated for the Mu Us Desert. (a) Land cover distribution over the study area; (b)-(g) annual average SM from 2015-2020; (h)-(j) annual precipitation and annual average temperature of three sites (53529, 53723 and 53740), whose surroundings are mainly grassland, cultivated land, and cultivated land, respectively.

## 4. Discussion

### 4.1 Variable importance assessment

The selection of variables is an important step of a nonlinear regression model. The importance analysis of the variables carried out for this research found that a larger number of variables can improve the regression effect of these models. Due to the variables obtained in this study come from multiple data sources, their preprocessing may affect the construction of regression models and their relationship with SM. Moreover, variables collinearity and hyperparameters also affect the importance relationship of variables. Figure 13 shows the average importance scores of each variable for the RF and XGB models across all available days. The importance scores of different variables in the RF based model and the XGB based model

are similar. LST and surface albedo both affect surface energy exchange and partition. LST is an important variable in both models, which is consistent with the study of Zhao et al. (2018). NSDSI is the most sensitive soil moisture index compared to LSWI and NDWI, which was demonstrated in Yue et al. (2019). Topographical factors also exhibit importance on SM, especially elevation. NDVI is more sensitive to vagetation index than EVI and LAI. However, their effect was smaller than that of soil moisture index. It indicates that the SM inversion based only on LST and VI is inadequate. The influence of soil texture (sand, silt and clay) is relatively weak.

The standard deviation of the importance scores of each variable is shown with error bars in Fig. 13. Its changes are mainly affected by the samples used in the regression model and the temporal variations in surface variables. For static variables such as soil structure and topographic factors, the changes in their importance scores mainly depend on the number and the location of the samples. Figure 13 also shows that their standard deviation is relatively small. Compared with static variables, the standard deviation of the importance scores of dynamic variables is significantly larger, especially for LST and LAI. This indicates that it is not reliable to construct a single regression model for a long time series.

In general, the variable importance analysis suggests that the selected variables are suitable for the construction of the regression model. Moreover, choosing 16 days as a time period to build a regression model benefits from obtaining a sufficient number of samples.

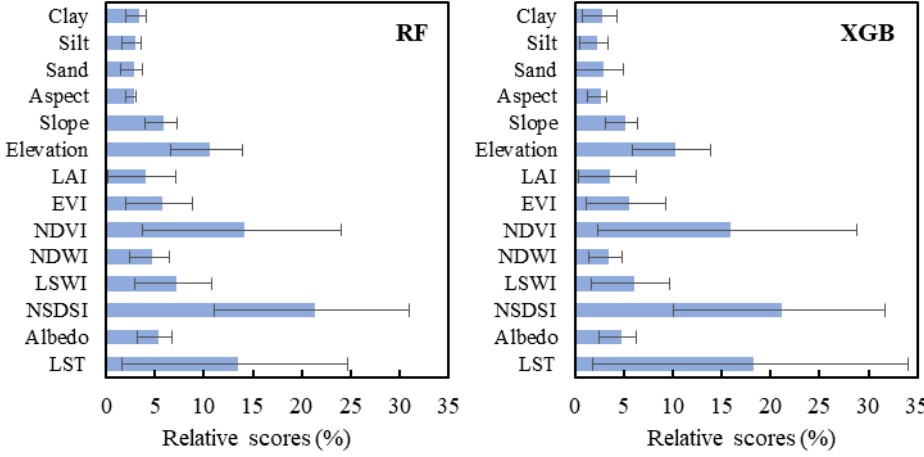

**Figure 13: The average importance scores of variables for the RF based approach and XGB based approach. Note: The importance scores are presented by increase in Node Purity (IncNodePurity) where the sum value is normalized for the RF model; The XGB model uses Gain to reflect the weight of variables.**

## 4.2 Advantages of model combination

Both RF and ANN have been applied to downscale remote sensed SM so far, especially RF (Zhao et al., 2018; Qu et al., 2019; Hu et al., 2020). This study showed that the simulation results of ANN have greater uncertainty, and the accuracy is generally worse than that of RF (Figs. 4 and 5). The RF algorithm shows a good simulation ability, but in comparison, the

XGB algorithm also has a corresponding effect or even higher. We also compared our simulation results combining multiple
models and the RF-based simulation results. The results showed that the combined products have higher accuracy than the RF-
based products, which is mainly reflected in the relatively more reasonable simulation of peaks and valleys (Table 4 and Fig
11). MLR has the worst effect compared to the other four models, which is likely to be affected by variable collinearity. In
fact, many algorithms, especially linear ones, exhibit more or less poor robustness when there is high collinearity between
variables (Dormann et al., 2013; Cammarota and Pinto, 2021). However, fewer explanatory variables often mean less ability
to explain target variables. Several studies have shown that ensemble tree algorithms such as RF and XGB are generally not
affected by variable collinearity (Tomaschek et al., 2018; Chen et al., 2020; Feng et al., 2021).
A combination of multiple methods can reduce overfitting and uncertainties for the simulation of long time series (Zanotti
et al., 2019; Yu et al., 2021). The five methods (MLR, SVR, ANN, RF and XGB) in this study have indicated the potential
flaws of a single model. Although the XGB model generally performs better than other models, it has still some shortcomings.
As it can be seen from Figs. 4 and 5, compared with the training accuracy, the test accuracy of the XGB model is significantly
reduced in several periods. This means that the simulation results of the XGB model is likely to have a certain degree of
overfitting. In contrast, the difference between training and test accuracy of the RF model is smaller. It showed better stability
than XGB at several periods (Figs. 4 and 5). The training accuracy of MLR and SVR has a small difference from the test
accuracy, but their overall accuracy is obviously lower (Table 4), which might be due to variable collinearity. Some studies
have proved that SVR may also perform better than ensemble algorithms (Yu et al., 2012; Fan et al., 2018). The fitting effect
of ANN varies greatly in different periods, indicating that its generalization is lower than other models (Piotrowski and
Napiorkowski, 2013). In general, the XGB and RF models provide the best combination of prediction accuracy and stability.
**4.3 Analysis of the relationship with precipitation and temperature**
Unlike predictors such as LST and NDVI that reflect SM status, climatic factors are are key drivers of SM variability. To
evaluate the impact of precipitation and temperature on SM, we performed a partial correlation analysis on the data of all
meteorological stations. Figure 14 shows that SM is mainly positively correlated with precipitation and temperature, and a few
regions are significantly negatively correlated with temperature. In terms of spatial distribution, SM of the sites in the eastern
region (including Inner Mongolia Province, Hebei Province and Shanxi Province) is mainly significantly affected by
precipitation. Due to the influence of glaciers and snowmelt, the SM of the sites in the western region (Xinjiang Province and
Gansu Province) is more affected by temperature. In addition, the number of sites with significant positive correlation with
precipitation and temperature is the largest in Qinghai Province. This indicates that precipitation and temperature in the eastern
part of the eastern Tibetan Plateau both have a great influence on SM.

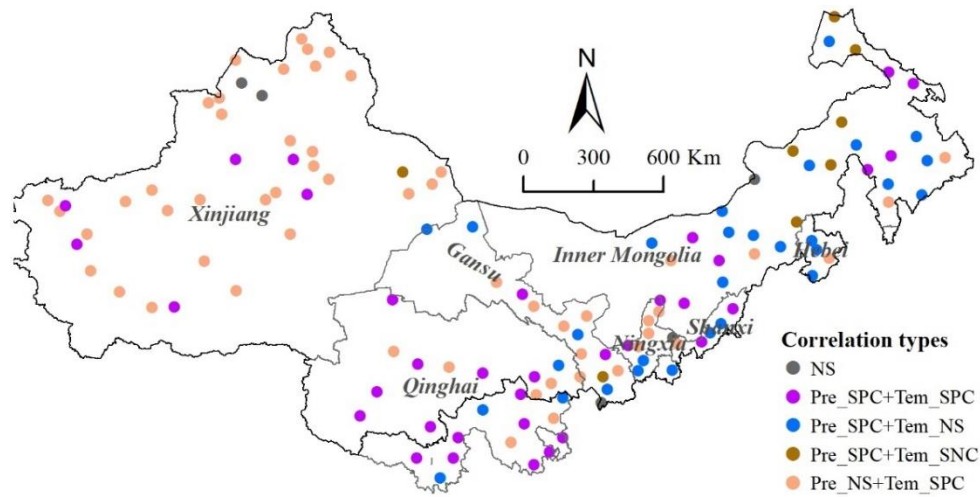


**Figure 14: Partial correlation between monthly downscaled SM and precipitation and temperature (Pre: precipitation; Tem:**
**temperature; NS: Not significance; SPC: Significantly positive correlation; SNC: Significantly negative correlation).**

## 4.4 Uncertainty and Prospects

While this study greatly improved the spatial resolution of SM data from 2015-2020 in the desertification areas of North
China by downscaling SMAP SM products, it still presents some shortcomings. Due to the influence of snow, ice and frozen
ground, the number of valid SMAP pixels during cold seasons is usually small, which limits the number of available samples.
With a period of 16-day, the number of valid samples may still be less than 100 during cold seasons (Fig. 3). The sample size
affects the simulation accuracy. Figures 4 and 5 show that there are seasonal variations in R and RMSE, which is likely to be
affected by the sample size. In general, a larger sample size often means more efficient sampling and more reliable results, but
not necessarily better evaluation accuracy. Likewise, insufficient samples can sometimes have good evaluation accuracy,
although the results are less reliable. In order to reduce the error caused by insufficient samples, this study replaced the periods
with less than 100 samples with the model of the previous periods. For this reason, the simulation results sometimes perform
poorly during cold seasons (Fig. 9). In addition, the upscaling (from 1 km to 36 km resolution) of surface variables also has a
certain impact on the accuracy of the model.
Our products have a good correlation with the in situ observation data. However, in situ observed SM data are limited in
their representation of the entire 1 km ×1 km grid. Figure 6 shows that the evaluation accuracy of different points varies greatly.
It indicates that the relationship between in situ observation data and remote sensing SM has great uncertainty due to the
influence of scale, and the same conclusion can also be found in some related studies (Zeng et al., 2015; Abbaszadeh et al.,
2019; Bai et al., 2019; Liu et al., 2019; Zhang et al., 2020). In addition, due to instrument accuracy and climate change, there
are some errors in the in situ observation data, especially at low temperatures. The in situ observed SM data obtained in this
paper are relatively limited, and their spatial distribution is concentrated in a certain part of of the study area, which is weak
representative. In order to verify the accuracy of the data as much as possible, this study also selected several sets of gridded
SM products for comparison. The results showed that our products perform better in temporal variability and spatial patterns
(Figs. 9 and 11).
**5 Data availability**
The codes mainly used in this paper mainly includes sample selection, the building of the optimal regression model and
the result prediction. The downscaled daily SM dataset at 1 km spatial resolution is available at
https://doi.org/10.6084/m9.figshare.16430478.v6 (Rao et al., 2022). The data maps are all provided in Geotiff format, and the
value has expanded 10,000 times to make them easier to store. The filenames reflect the production date in Julian Day format.
**6 Conclusions**
In this study, an approach was proposed for downscaling 36 km SMAP SM products using MODIS optical products and
other surface variables (mainly topographic data and soil data) based on multiple machine learning methods. Overall, the
regression performance of the five methods is, in order: XGB>RF>SVR>ANN>MLR. Compared with MLR, SVR and ANN,
XGB and RF have much better accuracy, and they were used in combination to produce daily 1 km downscaled SM in a period
of 16 days. The validation shows that the downscaled SM data are highly related to most in situ measured SM. The ubRMSE
with an average of 0.057 $m^3/m^3$ is generally less than 0.090 $m^3/m^3$ at the Maqu Network. Time series of SM data from in situ
observation sites were also compared. The results show that the downscaled SMs are highly related to SMAP SMs, and provide
a more complete time series and match better with the in situ measured SM. Compared with some commonly used gridded
SM products such as SMAP L2 (l km or 3 km), GCOMW/ASMR2, C3S, ERA5 and FLDAS SMs, the downscaled SM data
not only have higher spatial resolution, but also have a more reliable accuracy whether in time series or spatial distribution.
The maps of downscaled SM show larger values from June to September, which coincides with the vegetation growing
season. The difference in annual mean SM is small. Spatially, SM is relatively large in Qinghai Province and in northeastern
Inner Mongolia, especially in summer. In arid areas such as the Tarim Basin, SM is relatively small throughout the year.
Moreover, precipitation and temperature both have great influence on SM in the study area. Precipitation has a greater impact
on SM in the eastern part of the study area, while the effect of temperature appears to be more pronounced in the west.
This approach makes it possible to more accurately assess the soil moisture status in the study area. The results can support
regional agricultural planting and revegetation efforts and can be applied to limit desertification in other areas in the future.

**Author contributions.** FW and PR designed the research, developed the methodology, performed the analysis, and wrote the
paper; YW, YL, XW, and ZW edited and revised the paper.

**Competing interests.** The authors declare that they have no conflict of interest.

**Acknowledgements.** This work was supported by the National Key Research and Development Program of China (2018YFC0408103), the National Pilot Project for Ecological Protection and Restoration of Mountains, Rivers, Forests, Farmlands, Lakes and Grasslands (Grant No. WR0203A552018), and the Desertification Monitoring Project of National Forestry and Grass Administration (Grant No. 2020062012). We thank all data providers and the anonymous reviewers for their detailed and constructive comments.

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
