# Peer review of "Daily soil moisture mapping at 1 km resolution based on SMAP data"

_Earth System Science Data, 2021_

## Community Comment (CC2)

Dear commenter,

Thank you for your comments and suggestions. Based on your suggestion, I have made certain revisions to the manuscript. The first is to describe the XGB algorithm in more detail. The second is to discuss the deficiencies of the RF algorithm in more detail. The main revisions include the following:

(1) Introduction: Added description of shortcomings of RF algorithms

70  correlated with the dependent variable (Elshorbagy and Parasuraman, 2008; Ågren et al., 2021). Since RF algorithm uses
71  random sampling with replacement, its simulation results will not exceed the range of training set and tend to ignore some
72  extreme values when used as a regression model (Belgiu and Drăguț, 2016).  –Extreme gradient boosting (XGB), as a new
73  ensemble learning method (Chen and Guestrin, 2016), performs well in some fields (Wang et al., 2020; Fan et al., 2021; Ma
74  et al., 2021), but it has rarely been used for soil moisture downscaling. Compared with methods such as RF, XGB adopts the
75  boosting weighted sampling method, which can better simulate the extreme values existing in the samples (Chen and Guestrin,
76  2016). The coarse-resolution remote sensed SM (>10 km) itself has ignored some maxima or minima with relatively finer-grid
77  SMs, so a method that better simulates extreme values will obviously have certain theoretical advantages.

(2) Methods: Added the description of the main formula of the XGB algorithm

RF and XGB are tree based ensemble algorithms, which have prediction accuracy and good generalization ability, and are not prone to overfitting (Rao et al., 2018; Abbaszadeh et al., 2019). RF is a multiple-tree algorithm improved by  bootstrap to reduce decision tree bias in determining the splits (Mohana et al., 2021). Many studies have used RF to build regression models of remotely sensed SM and related variables, and almost all achieved better results compared to other regression methods (Zhao et al., 2018; Qu et al., 2019; Hu et al., 2020). In contrast, the application of XGB, which applies a

8

regularized gradient boosting framework, is still very limited. However, XGB has incomparable advantages in generalization performance and accuracy (Wang et al., 2020).

The XGB algorithm is a boosting-type ensemble of multiple CART decision trees (Chen and Guestrin, 2016). The predicted result of the boosting-type tree ensemble model can be expressed as follows:

$$\hat{y}_i = \emptyset(x_i) = \sum_{k=1}^{K} f_k(x_i), f_k \in F \qquad (4)$$

where $F$ is the space of regression tree, K is the total number of trees, which means the model uses K additive functions, $f_k(x_i)$ is the weighted score of the $k$-th tree on $i$-th input data $(x_i)$.

XGB adopts a regularized learning objective to optimize the simulation results.

$$Obj(\emptyset) = \sum_{i=1}^{N} l(y_i, \hat{y}_i) + \sum_{k=1}^{K} \Omega(f_k) \qquad (5)$$

where $l$ is the loss function, N is the total number of input, $\Omega$ is the regularization term to penalize the model complexity and prevent overfitting.

(3) Result 1: Added the comparison results using boxplots for all SM products and in situ observed SM.

[Figure]

Further, all gridded SM products are compared with in situ SM. Figure. 8 shows a significantly higher correlation between
the downscaled SM and in situ SM of the Maqu Network. The ubRMSE median of the downscaled SM is the smallest, and its
RMSE is second only to the C3S (0.25°) product. The bias of the downscaled SM is higher than that of some products, even
higher than the original SMAP L3 (36 km) data. Almost the same results can be obtained from in situ observations of Babao
Network. (Fig. S2). The difference is that the bias of the downscaled SM is lower than the result of SMAP L3 (36 km).
Compared with RF-based downscaled SM, the downscaled SM with multiple machine learning approaches performed better,
especially R and RMSE.

295
296    Figure 8: Comparison of gridded products and in situ observation SM of the Maqu Network.

(4) Result 2: Given that random forests have proven good methods in some literature and our computational results, we added the RF-based downscaled SM in Figures 9 and 11 (Figures 8 and 10 of the original manuscript).

[Figure]

[Figure]

[Figure]

(5) Discussion: Based on the simulation results of random forests, the advantages of the joint approachs are discussed.

In order to reduce the error caused by the method, we also adopt a combination of multiple models to obtain the optimal results. Both RF and ANN have been applied to downscale remote sensed SM so far, especially RF (Zhao et al., 2018; Qu et al., 2019; Hu et al., 2020). This study showed that the simulation results of ANN have greater uncertainty, and the accuracy is generally worse than that of RF (Figs. 4 and 5). The RF algorithm shows a good simulation ability, but in comparison, the XGB algorithm also has a corresponding effect or even higher. We also compared our simulation results combining multiple models and the RF-based simulation results. The results showed that the combined products have higher accuracy than the RF-based products, which is mainly reflected in the relatively more reasonable simulation of peaks and valleys (Figs. 9 and 11).

Looking forward to your next suggestions. Thank you!

---

## Author Comment (AC1)

Dear commenter,

Thank you for your comments and suggestions. Firstly, uncertainty. In order to improve the simulation accuracy, I have tried many machine learning methods in the early stage, and chose a more representative method to describe in the manuscript. And I also found that tree classifiers, especially Xgboost, have significantly better performance and efficiency compared to other classifiers.

Secondly, in order to improve the accuracy, we set 16 days as a regression period. It reduces errors caused by poor timeliness of dynamic variables (such as NDVI and EVI) and little valid data for one day.

Lastly, in order to verify the accuracy of our data, in addition to the limited in-situ observed soil moisture data and precipitation data, we also compared the data with some mainly existing reliable gridded soil moisture product, such as SMAP L2 SM (1 km and 3 km), GCOMW/ASMR2 SM (0.1°), C3S SM (0.25°), ERA5 SM (0.1°) and FLDAS SM (0.1°). It turns out that the data we produced has obvious advantages, which are mainly reflected in three points. First, the value of our data is generally in the middle of these products, and it is also relatively close to the in-situ measured values (see Figure 8). The second is time series. The product we produced generally has more valid data compared to other products, and its variation range is more reasonable than several other products (see Figure 8). The third is the spatial distribution of these products. Our products present a better spatial pattern of soil moisture, which is close to the actual situation, and its high spatial resolution makes some information displayed more clearly than other products (see Figure 10). Of course, the description of uncertainty in the manuscript is not detailed enough. We will try to modify this part of the content.

DOY is the day of year. Since all MODIS products use this Julian date, this manuscript also names the data in this way for convenience. This dataset is freely available at https://doi.org/10.6084/M9.FIGSHARE.16430478.V5. I will describe it in detail in the manuscript.

Looking forward to your next suggestions. Thank you!

[Figure]

**Figure 8: Time series of the in situ observed SM, the downscaled SM, the acquired gridded SM products and daily precipitation at the four selected SM sites (From Maqu Network and Babao Network, respectively).**

[Figure]

**Figure 10: Daily average SM from 2015-2020 in the study area. (a)-(f) are the downscaled SM (1 km), SMAP L3 SM (36 km), GCOMW/ASMR2 SM (0.1°), C3S SM (0.25°), ERA5 SM (0.1°) and FLDAS SM (0.1°), respectively.**

model. The in situ observed SM data obtained in this paper are relatively limited, and their spatial distribution is concentrated in a certain part of of the study area, which is not representative. It increases the uncertainty of the simulation results. In order to verify the accuracy of the data as much as possible, we compared this product with the existing more reliable gridded SM products, and the results showed that our product showed certain advantages in both time series and spatial distribution (Figs .8 and 10).

The Chinese government focuses on desertification reduction through afforestation and the establishment of grasslands. SM data with high temporal and spatial resolution can provide a reference for the next steps of revegetation.

**5 Code and data availability**

---

## Author Comment (AC2)

Dear reviewer,

Thank you for your comments and suggestions. Based on your suggestion, I have made certain revisions to the manuscript.

(1) misuse of desertification, monsoon and other geographic terms throughout the manuscript. The study region, defined by the authors, is not "areas affected by desertification", neither "monsoon climate region". Please check in detail.

Thank you very much for this suggestion. First, for the "monsoon climate region". I found that my description was wrong after submitting the manuscript. It should be a temperate continental climate.

Northern China is mostly arid with an annual precipitation of less than 400 mm. The region belongs to the temperate continental  climate and is subject to large-scale desertification. The desert areas of Northern China are susceptible

Second, for "areas affected by desertification". Our study area is provided by the National Forestry and Grassland Administration (China). I checked with the appropriate manager recently, and they thought that it is more scientific to replace "areas affected by desertification" with "desertification areas". All of these words have been changed in the manuscript.

**Daily soil moisture mapping at 1 km resolution based on SMAP data for desertification areas in Northern China**

(2) Line 18-19: are you sure "very sensitive to SM"?

Thank you very much for this suggestion. The ultimate purpose of our study is vegetation restoration in the study area, where soil moisture is a key indicator. "very sensitive to SM " is a bit absolute, I have modified it to " sensitive to SM".

such as vegetation index and surface temperature. Areas affected by desertification in Northern China, which are  sensitive to SM, were selected as the study area, and the downscaled SM with a resolution of 1 km on a daily scale from 2015 to 2020

(3) Line 32: provide references for GLDAS.

Related literature has been added.

In the past, SM data were mainly obtained through ground measurements or the assimilation of products based on land surface models such as the Global Land Data Assimilation System (GLDAS) (Fang and Lakshmi, 2014; Zawadzki and Kędzior, 2016; Liu et al., 2021). Although most accurate SM data at different soil depths can be obtained, field measurements and in

(4) Line 36-40: data assimilation products may be produced with satellite data as inputs. Thus, it is not independent on remote sensing. Modify your statements.

I'm afraid I did not make it clearly. What I want to express in this sentence is: the accuracy of the soil moisture product of data assimilation is mainly affected by the land surface model and the original input data. The remote sensing data you mentioned is the original input data. It is not the same as the remote sensing SM product we mentioned above. In order to avoid ambiguity, I have made some modifications.

(Peng et al., 2021). Compared to ground measurements, remote sensing products can provide good spatial and temporal coverage of SM with a relatively low cost to the user (Zeng et al., 2015; Zhao et al., 2018; Meng et al., 2020). Data assimilation SM products largely depend on the accuracy of the land surface model and the original inputs data (Zawadzki and Kedzior,

(5) Line 44: what does the "very stable" mean here? Passive microwave radiometer data are sensitive to more influences, such as atmospheric effects and surface vegetation.

I'm afraid I did not make it clearly. I would have liked to express that passive remote sensing products are generally more stable compared to active remote sensing. In order to avoid ambiguity, I deleted it.

scattering and greatly affected by the surface roughness and vegetation types (Lievens et al., 2011; Wagner et al., 2013). Unlike active sensors, passive microwave radiometers or sensors are rarely affected by scattering have almost no scattering and generate very stable SM products (Abbaszadeh et al., 2019). Common passive microwave SM products are listed in Table 1

(6) Line 56: what does 'directly retrieve' mean?

I'm afraid I did not make it clearly. It has been modified.

proposed the temperature vegetation dryness index (TVDI) and used it to assess the SM status. Despite their higher resolution, however, optical remote sensing data do not allow to directly retrieve true SM. Relative SM indicators can be calculated using optical remote sensing data, however, reliable ground measurements or other data are still required to obtain the true value of SM.

(7) Each method produces a dataset. That does not mean the multiple machine learning methods produce the datasets following the normal distribution. In this sense, statistical mean may be biased, which is well-known to climate community.

Thank you very much for this suggestion. Our study uses a combination of multiple machine learning to select the best regression model for each period and not by taking an average for the SM result.

(8) Line 91-92: wrong description of the region with "monsoon climate". So is the desertification.

Thank you very much for your suggestions. I have made modifications. Refere to Question (1).

(9) Line 93: "water-vapor-ecosystem", what does it mean?

I'm sorry for the non-standard expression. I made a simple modification.

to climate and hydrological changes and have fragile ecosystems. Soil water is a key parameter in land-atmosphere interactions(Ma et al., 2019), and its change greatly affects the survival of vegetation and

(10) Line 115: Give the full spelling for NDWI, LSW, ECMWF, EVI, geotiff and many others for their first appearance in text.

Thank you very much for this suggestion. I have carefully checked all abbreviations, and some abbreviations that do not appear are added with full spelling.

The soil wetness related indexes, including NDWI, NSDSI, and Land Surface Water Index (LSWI), were produced using bands of the MOD09A1 product. Their formulas are:

160  reanalysis dataset (ERA5) produced by European Centre for Medium-Range Weather Forecasts (ECMWF) provides

(11) The parameters used for ML are linearly correlated. Does it affect your results?

Thank you very much for this suggestion. Collinearity between variables will affect the simulation results, which is not considered in the description process of this paper. We add some content in Section 4.2.

In general, except for ensemble algorithms (including RF and XGB), collinearity is more or less affected. Due to this advantage of the ensemble algorithm, many studies generally do not consider multicollinearity problems when using random forests for regression or classification.

389    **4.2 Advantages of model combination**

390        Both RF and ANN have been applied to downscale remote sensed SM so far, especially RF (Zhao et al., 2018; Qu et al.,
391    2019; Hu et al., 2020). This study showed that the simulation results of ANN have greater uncertainty, and the accuracy is
392    generally worse than that of RF (Figs. 4 and 5). The RF algorithm shows a good simulation ability, but in comparison, the
393    XGB algorithm also has a corresponding effect or even higher. We also compared our simulation results combining multiple

394    models and the RF-based simulation results. The results showed that the combined products have higher accuracy than the RF-
395    based products, which is mainly reflected in the relatively more reasonable simulation of peaks and valleys (Figs. 9 and 11).
396    MLR has the worst effect compared to the other four models, which is likely to be affected by variable collinearity. In fact,
397    many algorithms, especially linear ones, exhibit more or less poor robustness when there is high collinearity between variables
398    (Dormann et al., 2013; Cammarota and Pinto, 2021). However, several studies have shown that ensemble algorithms such as
399    RF and XGB are generally not affected by multicollinearity (Tomaschek et al., 2018; Chen et al., 2020; Feng, 2021).

(12) Line 177: incomparable?

This word I refer to the literature (Wang et al., 2020). The expression is a bit absolute, replaced by "prominent".

Extreme Gradient Boosting (XGBoost), as a new ensemble learning method, was proposed by Dr. Chen Tianqi at the university of Washington in 2016 [25]. For now, this method has incomparable advantages in generalization performance, speed and accuracy compared with other ensemble learning algorithms [26–28]. For instance,

(13) Equations for RMSE (6) and (8) are wrongly expressed.

There should be nothing wrong with these two equations. The following is the equations from the literature (Hu et al., 2020).

$$RMSE = \sqrt{E\left[(\theta_{SMAP} - \theta_{insitu})^2\right]} \tag{7}$$

$$ubRMSE = \sqrt{E\left\{\left[(\theta_{SMAP} - E[\theta_{SMAP}]) - (\theta_{insitu} - E[\theta_{insitu}])\right]^2\right\}} \tag{8}$$

where $E[\bullet]$ represents the mean operator, $\theta_{isitu}$ is the in situ SM, $\theta_{SMAP}$ is the downscaled SM, $\sigma_{SMAP}$ is the standard deviation of SMAP SM, and $\sigma_{SMAP}$ is the standard deviation of the in situ SM.

(14) Figures 4 and 5: there are clearly seasonal variation in correlation coefficient and RMSE. It means significant systematic errors in the products. Give scientific explanation to the data reliability.

Thank you very much for your suggestion. Since the overall simulation effect is better, I have not considered this issue carefully before. My understanding is that the relatively poor effect in summer may be due to the effect of noise such as clouds. I added it in Section 4.1 (Discussion part).

While this study greatly improved the spatial resolution of SM data from 2015-2020 in the desertification areas of North China by downscaling SMAP SM products, it still presents some shortcomings. Although we chose high-quality MODIS images, variables such as daily LST and Albedo are still affected by clouds. It will have some impact on the simulation results, especially in the rainy season. Figures 4 and 5 show that the simulation results are generally better in the cold season with less clouds, and worse in summer with more clouds. For example, Ddue to the image quality and coverage of SMAP and the impact of noise from clouds on the MODIS products, the number of valid samples for a 16-day period may still be less than 100 points in cold seasons (Fig. 3). This study replaced the periods with less than 100 samples with the model of the previous periods.

(15) Line 251-260: The errors are large between the Maqu and the Bbaso network, which need substantial investigation.

The evaluation accuracy of Babao network is generally lower than that of Maqu Network, the first reason is the measured soil depth. The soil depth measured by SMAP is 5 cm, the same as that measured by Maqu Network, while that measured by Babao Network is 4 cm. Another reason is that there is a bias between site measurements and remote sensing data itself. The relationship between the 1 km×1 km grid and the site itself will have a large error. In this study, even SM of some sites from Maqu network

did not match well the Remote sensing SM. To make the results more convincing, I added the following.

291   Further, all gridded SM products are compared with in situ SM. Figure. 8 shows a significantly higher correlation between
292 the downscaled SM and in situ SM of the Maqu Network. The ubRMSE median of the downscaled SM is the smallest, and its
293 RMSE is second only to the C3S (0.25°) product. The bias of the downscaled SM is higher than that of some products, even
294 higher than the original SMAP L3 (36 km) data. Almost the same results can be obtained from in situ observations of Babao
295 Network. (Fig. S2). The difference is that the bias of the downscaled SM is lower than the result of SMAP L3 (36 km).
296 Compared with RF-based downscaled SM, the downscaled SM with multiple machine learning approaches performed better,
297 especially R and RMSE.
298

[Figure]

299
300   Figure 8: Comparison of gridded products and in situ observation SM of the Maqu Network.

[Figure]

Figure S2: Comparison of gridded products and in situ observation SM of the Babao Network.

(16) Line 283: "due to spatial resolution" is a superficial reasoning. Insightful clarification should be given.

I'm afraid I didn't express clearly. This sentence seems more like a discussion, so I deleted it. For a detailed explanation, refer to Section 4.4.

with the precipitation.

. The variation trends of the RF-based

(17) Line 291: here appears 'process of vegetation growth'. SMAP SM data are subject to vegetation cover, which is known in the field, but the authors failed to address it.

This study does not consider the quality of SMAP data itself. Generally speaking, passive remote sensing is relatively less affected by surface roughness and vegetation. On the contrary, active remote sensing is more affected.

In fact, I add this sentence here mainly to express that the high SM from April to September is consistent with the seasonal law of vegetation growth. In order not to cause ambiguity, I removed it.

SM varies greatly in different months in desertified areas. Figure 10 shows the average SM in each month in the study area. The SM shows a monthly change pattern, and the values from June to September are bigger than in other months, especially in southern Qinghai Province, eastern Inner Mongolia Province, and western Xinjiang Province. The SM in some areas is low throughout the year, such as in the Tarim Basin of Xinjiang

(18) Line 295: "little variation"? change the words.

Thanks for your advice! It has been modified.

The annual average SM was also calculated (Fig. S3). Compared with the monthly average SM, the annual average SM changed significantly less.. Further, we compared the spatial

(19) There are too many "some" in text. Vague expression.

Thank you very much for your suggestion. The unnecessary "some" of the manuscript have been deleted. Some expressions are also further modified.

(20) Line 327: strange subtitle.

I referred to this writing form Zhao et al. (2018). Not very scientific, I modified it to "Variable importance assessment".

**4. Discussion**

**4.1 Variable importance assessment**

(21) Line 335: "influence of soil texture (sand, silt and clay) is relatively weak, but it cannot be completely ignored.". why?

It caused some ambiguity and I deleted the latter part.

NDWI, which was demonstrated in Yue et al. (2019). Topographical factors also exhibit importance on SM, especially elevation. The influence of soil texture (sand, silt and clay) is relatively weak.

(22) Line 347: IncNodePurity? What is it?

It reflects an important indicator, and I added its full spelling.

Figure 13: The average importance scores of variables for the RF based approach and XGB based approach. Note: The importance scores are presented by increase in Node Purity (IncNodePurity) where the sum value is normalized for the RF model; The XGB model uses Gain to reflect the weight of variables.

(23) Line 350: various noises? How many?

My expression was not clear, deleted "various".

The simulation results of long time series will inevitably suffer the interference of various noises. A combination of multiple methods can reduce overfitting and uncertainties (Zanotti et al., 2019; Yu et al., 2021). The five methods (MLR, SVR,

(24) Line 367: mainly significantly. Remain one.

I'm afraid I didn't express clearly. "mainly" was modified to "more"

SM is mainly positively correlated with precipitation and temperature, and a few regions are significantly negatively correlated with temperature. In terms of spatial distribution, SM of the sites in the eastern region (including Inner Mongolia Province, Hebei Province and Shanxi Province) is mainly more significantly affected by precipitation. Due to the influence of glaciers and snowmelt, the SM of the sites in the western region (Xinjiang Province and Gansu Province) is more affected by

(25) Line 382-383: delete it.

It has been removed.

(26) Line 391: "a framework was proposed"? It does not make sense.

I'm afraid I didn't express clearly. "a framework" was modified to "an approach".

In this study, a framework an approach was proposed for downscaling 36 km SMAP SM products using MODIS optical products and other surface variables (mainly topographic data and soil data) based on multiple machine learning methods.

Finally, thanks a lot for your careful review and invaluable advices. Looking forward to the opportunity to learn from you! I also made some other revisions, please refer to other review results.

Looking forward to your next suggestions. Thank you!

---

## Author Comment (AC3)

Dear editor,

Thank you for your comments and suggestions. First of all, we are so sorry for taking so long time to reply to you. We are also aware of many problems in our manuscript. We have tried our best to revise our manuscript according to the comments. Attached please find the revised version, which we would like to submit for your kind consideration. We have written a point-by-point response letter for all reviewers. To make the reply more visible, Q represents questions raised by reviewers, and A are our answers for these questions. Below are your questions and suggestions:

(1) 69 there is no such thing as "inefficient samples". Do you mean "insufficient samples"?

A1: Thank you for your suggestion. It should be "inefficient sampling". It mainly refers to less scientific or reasonable sample sets. (Line 74)

(2) 83 unclear expression "Considering the role of SM in the ecological environment" -- "for ecology" or "for the environment"

A2: Sorry for the unclear expression. The "ecological environment" was changed as "terrestrial ecosystems". (Line 95)

(3) l. 96 add reference to Figure 1. All figures must be referred to in the text. (this reference only appears in line 136)

A3: Thank you very much for your suggestions. We have added it in the Section 2.1 (Lines 109 and 110). Please see the revised version.

(4) general comment, page 4: it would be helpful to add a figure showing the precipitation climatology of the region and its variability rather than only showing the topography. This would allow readers to evaluate the geographic patterns of the SM results.

A4: Thank you very much for your suggestions. A figure showing the annual mean precipitation is added in Figure 1. Please see the revised version.

(5) 144 following: to facilitate comparison of dataset resolutions, please convert all resolutions to km. Example: "It has a spatial resolution of 0.25 degrees (~28 km)". Also use the degree symbol consistently.

A5: Thank you very much for your suggestions. This modification is implemented in the full text. Please see the revised version.

(6) sections 2.2.2 to 2.2.4: please include more details about the datasets, such as spatial resolution, time span covered and hints about the evaluation of their quality.

A6: Thank you very much for your suggestions. We added some content about these products, and the temporal and spatial resolutions of these products are described in detail. Please see the revised version.

(7) Figure 2: what is a "set dataset"? I only know the terms "validation dataset" and "test dataset"

A7: Thank you very much for your suggestions. It is described wrong, and it should be training set and test set. We have modified it. Please see the revised version.

(8) section 2.3.2: please define the target data product more more clearly. The section is named "downscaling", but the text describes a coarsening of resolution (to 36 km) and a functional mapping/regression from predictor variables to the predictant. What is the output resolution and how is this *downscaled*?

A8: Thank you very much for your suggestions. The downscaling process is described in more detail in the revised version. The process is divided into: the construction of 16-day regression model, prediction of 1-km daily SM product. Please refer to the revised version.

(9) 202: how was the data split done? Given that there is substantial memory in the system and some data are used with a temporal resolution of 16 days, a simple random sampling approach is invalid as it will lead to overoptimistic validation results. This issue is discussed for example in Schultz et al., 2021, https://royalsocietypublishing.org/doi/10.1098/rsta.2020.0097.

A9: Thank you very much for your suggestions. Based on the previous research (https://doi.org/10.1016/j.rse.2014.02.015; https://doi.org/10.1016/j.rse.2019.111261; https://doi.org/10.1016/j.rse.2021.112646 ), we have improved this work. Stratified sampling considering the date of sample acquisition within 16 days was applied to split the training and test sets which means the number of training and test samples taken each day is equal. In addition, to ensure that the training and test sets of stratified sampling do not differ too much, the Kolmogorov-Smirnov (KS) test is adopted to test the distribution consistency of them. Based on this new samples, the model was re-run, and the results throughout the article were subject to change. Please see the revised manuscript for all revisions.

(10) section 3.2: please add more quantitative evaluation results. There are many statements like "good" or "poor", but it remains unclear according to which target criterion these quality indicators are given. For example, what is an acceptable R2 value in your view and why?

A10: Thank you very much for your suggestions. We have made a lot of modifications to Section 3.2. First, more quantitative evaluation results was added (Line 317). Second, in order to verify more scientifically, a figure (Figure 8: Comparison of gridded products and in situ observation SM of the Maqu Network) was added, and it shows that our simulation results have better accuracy compared to other products. Third, about R, we added the content "R greater than 0.6 and 0.8 indicates there is a reliable and strong correlation (Akoglu, 2018)" in Section 3.1 (Line 322). Fourth, there are some problems with the Figure 5, and The R values of the test set are not all greater than 0.8. The new results are all from new simulation results.

(11) l. 303: language. What does this mean "The average SM of the ERA5 products is polarised"?

A11: Sorry for my unscientific expression. What we mainly want to express is that: The average SM is significantly overestimated in the southern part of the study area, and underestimated in some areas in the northern of the study area. We have made some modifications (Line 431).

(12) section 4.1: be aware that there is a fine difference between the influence of a physical quantity on the target variable and the importance of input variables for the model regression. The latter can depend on the model architecture, data preprocessing and normalisation and other factors which have nothing to do with the cause-effect relationships in the real world. The text is ok as is, but it may help readers to assess the relevance of your results if you alert them to this point.

A12: Thank you very much for your suggestions. Due to the variables obtained in this study come from multiple data sources, data preprocessing can make errors, which I have added it in section 4.1 (Line 473).

(13) l. 350: The sentence "The simulation results of long time series will inevitably suffer the interference of various noises." is meaningless and not based on a sound mathematical concept from statistics.

A13: Thank you very much for your suggestions. It has been modified.

(14) l. 356 remove "even" before "smaller"

A14: Thank you very much for your suggestions. It has been deleted.

(15) general comment on section 4.2 and also methods section: your statements about the quality of individual ML models are not supported by the evidence shown in the paper.

A15: Thank you very much for your suggestions. We have made some modifications. The results of RF and XGB were compared with the combined result (Table 4 and Figure 8), and the combined result is better compared to a single model. Please see the revised manuscript for all revisions.

(16) Information is missing about the detailed set-up of these ML models. Otherwise you might be comparing apples and oranges, for example if a very small ANN is compared to an RF with many trees and branches. Have you done any ablation studies to find the optimum set-up for each ML method? Furthermore, the discussion would benefit from a somewhat more elaborate reflection about why a certain ML model performs better or worse for certain seasons, groups of stations, etc.

A16: Thank you very much for your suggestions. We tuned the hyperparameters in the process of doing this work, but this manuscript did not describe in detail. In the new version, we describe the parameter tuning process in more detail. Please refer to Attachment (Part 2: Hyperparameter Optimization Results).

In addition, regarding the difference in seasons, we think that it is mainly affected by the sample size (Line 344), and we also added discussion in the new version (Line 529).

(17) section 4.3 How do these results relate to the variable importance analysis presented in Figure 12?

The variable importance analysis is mainly for the predictors, and their can reflect the status of SM. Meteorological factors are the driving factors for SM. We have expressions in the new version. (Line 513)

Finally, thanks a lot for your careful review and invaluable advices. The version I trust again can meet your requirements. If that is not enough, seven of your suggestions; if possible, I am eager for this manuscript to be re-reviewed. Looking forward to the opportunity to learn from you! I also made some other revisions, please refer to other review results.

Looking forward to your next suggestions. Thank you!

Sincerelly!

Pinzeng Rao

Address: Beijing, China.

Email: 578264905@qq.com

---

## Author Comment (AC4)

Dear Reviewer #1 (RC1)

Thank you for your comments and suggestions. Based on your suggestion, I have made certain revisions to the manuscript. We will answer your question is divided into several sections:

Uncertainty. In order to improve the simulation accuracy, we have tried many machine learning methods in the early stage, and chose a more representative method to describe in the manuscript. And we also found that tree classifiers, especially XGB, have significantly better performance and efficiency compared to other classifiers.

In order to improve the accuracy, we set 16 days as a regression period. It reduces errors caused by poor timeliness of dynamic variables (such as NDVI and EVI) and little valid data for one day.

In order to verify the accuracy of our data, in addition to the limited in-situ observed soil moisture data and precipitation data, we also compared the data with some mainly existing reliable gridded soil moisture product, such as SMAP L2 SM (1 km and 3 km), GCOMW/ASMR2 SM (0.1°), C3S SM (0.25°), ERA5 SM (0.1°) and FLDAS SM (0.1°). It turns out that the data we produced has obvious advantages, which are mainly reflected in three points. First, the value of our data is generally in the middle of these products, and it is also relatively close to the in-situ measured values (see Figure 8). The second is time series. The product we produced generally has more valid data compared to other products, and its variation range is more reasonable than several other products (see Figure 9). The third is the spatial distribution of these products. Our products present a better spatial pattern of soil moisture, which is close to the actual situation, and its high spatial resolution makes some information displayed more clearly than other products (see Figure 11). Of course, the description of uncertainty in the manuscript is not detailed enough. We will try to modify this part of the content.

DOY is the day of year. Since all MODIS products use this Julian date, this manuscript also names the data in this way for convenience. This dataset is freely available at https://doi.org/10.6084/M9.FIGSHARE.16430478.V5. We will describe it in detail in the manuscript.

In addition to what has been described above, we have also made extensive revisions to the paper. For example, the comparison of grid SM products with in situ data was added to verify the accuracy of our products. Please see the modified version.

Looking forward to your next suggestions. Thank you!

Sincerelly!

Pinzeng Rao

Address: Beijing, China.

Email: 578264905@qq.com

---

## Author Comment (AC5)

Dear Reviewer #2 (RC2)

Thank you for your comments and suggestions. Based on your suggestion, I have made certain revisions to the manuscript.

Q1: misuse of desertification, monsoon and other geographic terms throughout the manuscript. The study region, defined by the authors, is not "areas affected by desertification", neither "monsoon climate region". Please check in detail.

A1: Thank you very much for this suggestion. The study area is mostly "temperate continental climate", not"monsoon climate region". It was deleted. (Line 104)

For "areas affected by desertification". Our study area is provided by the National Forestry and Grassland Administration (China). We checked with the appropriate manager recently, and they thought that it is more scientific to replace "areas affected by desertification" with "desertification areas". All of these words have been changed in the manuscript. Please see the modified version.

Q2: Line 18-19: are you sure "very sensitive to SM"?

A2: Thank you very much for this suggestion. The ultimate purpose of our study is vegetation restoration in the study area, where soil moisture is a key indicator. "very sensitive to SM " is a bit absolute, we have modified it to " sensitive to SM". (Line 20)

Q3: Line 32: provide references for GLDAS.

A3: Thank you very much for this suggestion. Related literature has been added. (Line 33)

Q4: Line 36-40: data assimilation products may be produced with satellite data as inputs. Thus, it is not independent on remote sensing. Modify your statements.

A4: Thank you very much for this suggestion. In order to avoid ambiguity, we have made some modifications. (Line 40)

Q5: Line 44: what does the "very stable" mean here? Passive microwave radiometer data are sensitive to more influences, such as atmospheric effects and surface vegetation.

A5: Thank you very much for this suggestion. We would have liked to express that passive remote sensing products are generally more stable compared to active remote sensing. In order to avoid ambiguity, we deleted it.

Q6: Line 56: what does 'directly retrieve' mean?

A6: It has been modified (Line 59). Hope to help you understand.

Q7: Each method produces a dataset. That does not mean the multiple machine learning methods produce the datasets following the normal distribution. In this sense, statistical mean may be biased, which is well-known to climate community.

A7: Thank you very much for this suggestion. Our study uses a combination of multiple machine learning to select the best regression model for each period and not by taking an average for the SM result.

Q8: Line 91-92: wrong description of the region with "monsoon climate". So is the desertification.

A8: Thank you very much for your suggestions. We have made modifications. Refere to Question (1).

Q9: Line 93: "water-vapor-ecosystem", what does it mean?

A9: Thank you very much for your suggestions. It has been modified.(Line 105)

Q10: Line 115: Give the full spelling for NDWI, LSW, ECMWF, EVI, geotiff and many others for their first appearance in text.

A10: Thank you very much for your suggestions. We have carefully checked all abbreviations, and some abbreviations that do not appear are added with full spelling. Please see the modified version.

Q11: The parameters used for ML are linearly correlated. Does it affect your results?

A11: Thank you very much for your suggestions. Collinearity between variables will affect the simulation results, which is not considered in the description process of this paper. We add some content in Section 4.2. In general, except for ensemble algorithms (including RF and XGB), collinearity is more or less affected. Due to this advantage of the ensemble algorithm, many studies generally do not consider multicollinearity problems when using random forests for regression or classification. We have added discussion in section 4.2. Please see the modified version.

Q12: Line 177: incomparable?

Q12: Thank you very much for your suggestions. The expression is a bit absolute, replaced by "prominent". (Line 206)

Q13: Equations for RMSE (6) and (8) are wrongly expressed.

A13: Thank you very much for your suggestions. There should be nothing wrong with these two equations. The following is the equations from the literature (Hu et al., 2020).

$$\text{RMSE} = \sqrt{E\left[(\theta_{SMAP} - \theta_{insitu})^2\right]} \tag{7}$$

$$\text{ubRMSE} = \sqrt{E\left\{[(\theta_{SMAP} - E[\theta_{SMAP}]) - (\theta_{insitu} - E[\theta_{insitu}])]^2\right\}} \tag{8}$$

where $E[\bullet]$ represents the mean operator, $\theta_{isitu}$ is the in situ SM, $\theta_{SMAP}$ is the downscaled SM, $\sigma_{SMAP}$ is the standard deviation of SMAP SM, and $\sigma_{SMAP}$ is the standard deviation of the in situ SM.

Q14: Figures 4 and 5: there are clearly seasonal variation in correlation coefficient and RMSE. It means significant systematic errors in the products. Give scientific explanation to the data reliability.

A14: Thank you very much for your suggestions. Regarding the difference in seasons, we think that it is mainly affected by the sample size (Line 344), and we also added discussion in the new version (Line 539).

Q15: Line 251-260: The errors are large between the Maqu and the Bbaso network, which need substantial investigation.

A15: Thank you very much for your suggestions. The evaluation accuracy of Babao network is generally lower than that of Maqu Network, the first reason is the measured soil depth. The soil depth measured by SMAP is 5 cm, the same as that measured by Maqu Network, while that measured by Babao Network is 4 cm. Another reason is that there is a bias between site measurements and remote sensing data itself. The relationship between the 1 km×1 km grid and the site itself will have a large error. In this study, even SM of some sites from Maqu network did not match well the Remote sensing SM.

To make the results more convincing, we added some comparison between the grid SM data and the in-situ SM data. (Figure 8: Comparison of gridded products and in situ observation SM of the Maqu Network; Figure S2: Comparison of gridded products and in situ observation SM of the Babao Network). Please see the modified version.

Q16: Line 283: "due to spatial resolution" is a superficial reasoning. Insightful clarification should be given.

A16: Thank you very much for your suggestions. For a detailed discussion, please refer to the revised version (Section 4.4: Uncertainty)

Q17: Line 291: here appears 'process of vegetation growth'. SMAP SM data are subject to vegetation cover, which is known in the field, but the authors failed to address it.

A17: Thank you very much for your suggestions. It was deleted.

Q18: Line 295: "little variation"? change the words.

A18: Thanks for your advice! It has been modified. (Line 429)

Q19: There are too many "some" in text. Vague expression.

A19: Thank you very much for your suggestion. The unnecessary "some" of the manuscript have been deleted. Some expressions are also further modified.

Q20: Line 327: strange subtitle.

A20: Thank you very much for your suggestion. It is modified to "Variable importance assessment". (Line 469)

Q21: Line 335: "influence of soil texture (sand, silt and clay) is relatively weak, but it cannot be completely ignored.". why?

A21: Thank you very much for your suggestion. It caused some ambiguity and we deleted the latter part. (Line 424)

Q22: Line 347: IncNodePurity? What is it?

A22: Thank you very much for your suggestion. It reflects an important indicator, and we added its full spelling. (Line 436)

Q23: Line 350: various noises? How many?

A23: Thank you very much for your suggestion. My expression was not clear, we deleted this sentence. (Line 450)

Q24: Line 367: mainly significantly. Remain one.

A24: Thank you very much for your suggestion. "mainly" was modified to "more"

Q25: Line 382-383: delete it.

A25: Thank you very much for your suggestion. It has been removed.

Q26: Line 391: "a framework was proposed"? It does not make sense.

A26: Thank you very much for your suggestion. "a framework" was modified to "an approach".( Line 523)

Finally, thanks a lot for your careful review and invaluable advices. Looking forward to the opportunity to learn from you! I also made some other revisions, please refer to other review results.

Looking forward to your next suggestions. Thank you!

Sincerelly!

Pinzeng Rao

Address: Beijing, China.

Email: 578264905@qq.com